# Dual functioning by the PhoR sensor is a key determinant to *Mycobacterium tuberculosis* virulence

**Prabhat Ranjan Singh[1¤a], Harsh Goar[1☺¤b], Partha Paul[1☺], Khushboo Mehta[1,2], Bhanwar Bamniya[1,2], Anil Kumar Vijjamarri[1¤c], Roohi Bansal[1], Hina Khan[1], Subramanian Karthikeyan[1,2], Dibyendu Sarkar[1,2]***

1 CSIR-Institute of Microbial Technology, Sector 39 A, Chandigarh, India, 2 Academy of Scientific and Innovative Research (AcSIR), Ghaziabad, India

☺ These authors contributed equally to this work.
¤a Current address: Department of Microbiology and Immunology, Weil Cornell Medicine, New York, New York, United States of America
¤b Current address: Department of Medicine, Division of Hematology-Oncology UT Southwestern Medical Center, Dallas, Texas, United States of America
¤c Current address: National Institute of Child Health and Human Development, National Institutes of Health, Bethesda, Maryland, United States of America
* dibyendu@imtech.res.in

**Data Availability Statement:** All RNA sequencing data have been deposited in the GEO database with accession number GSE180161. All other relevant

## Abstract

PhoP-PhoR, one of the 12 two-component systems (TCSs) that empower *M. tuberculosis* to sense and adapt to diverse environmental conditions, remains essential for virulence, and therefore, represents a major target to develop novel anti-TB therapies. Although both PhoP and PhoR have been structurally characterized, the signal(s) that this TCS responds to remains unknown. Here, we show that PhoR is a sensor of acidic pH/high salt conditions, which subsequently activate PhoP via phosphorylation. In keeping with this, transcriptomic data uncover that acidic pH- inducible expression of PhoP regulon is significantly inhibited in a PhoR-deleted *M. tuberculosis*. Strikingly, a set of PhoP regulon genes displayed a low pH-dependent activation even in the absence of PhoR, suggesting the presence of non-canonical mechanism(s) of PhoP activation. Using genome-wide interaction-based screening coupled with phosphorylation assays, we identify a non-canonical mechanism of PhoP phosphorylation by the sensor kinase PrrB. To investigate how level of P~PhoP is regulated, we discovered that in addition to its kinase activity PhoR functions as a phosphatase of P~PhoP. Our subsequent results identify the motif/residues responsible for kinase/phosphatase dual functioning of PhoR. Collectively, these results uncover that contrasting kinase and phosphatase functions of PhoR determine the homeostatic mechanism of regulation of intra-mycobacterial P~PhoP which controls the final output of the PhoP regulon. Together, these results connect PhoR to pH-dependent activation of PhoP with downstream functioning of the regulator. Thus, PhoR plays a central role in mycobacterial adaptation to low pH conditions within the host macrophage phagosome, and a PhoR-deleted *M. tuberculosis* remains significantly attenuated in macrophages and animal models.

data are within the paper and its Supporting Information files.

**Funding:** This study received financial support from CSIR-IMTECH intramural grant OLP-0170, CSIR-funded FBR project MLP-0049, and SERB-funded project (EMR/2016/004904) to D.S. P.R.S was supported by DBT fellowship; H.G., P.P., K.M., B.B., A.K.V., and H.K. were supported by CSIR; R.B was supported by UGC fellowship. The funders had no role in study design, data collection and analysis, decision to publish or preparation of the manuscript.

**Competing interests:** The authors have declared that no competing interests exist.

## Author summary

Virulence-associated PhoP-PhoR of *M. tuberculosis* represents an attractive target to develop anti-tubercular therapy, but to date, the signal(s) that this regulatory system responds to remains unknown. We discovered that acidic pH and high salt conditions activate PhoP using a PhoR-dependent mechanism. Thus, pH inducible PhoP regulon expression is significantly impacted in a PhoR-depleted *M. tuberculosis* H37Rv. Our subsequent investigations reveal that homeostatic mechanism of regulation of P~PhoP relies on kinase/phosphatase dual functioning of PhoR, which determines mycobacterial pH homeostasis by controlling the final output of the PhoP regulon. Unexpectedly, global regulatory studies uncover that there can be PhoR-independent mechanism(s) of *in vivo* activation of PhoP. While probing for a non-canonical mechanism, we demonstrate that the sensor kinase (SK) PrrB phosphorylates PhoP. These results connect two SKs with signal-dependent activation of PhoP.

## Introduction

*M. tuberculosis* is highly adaptable to complex and varying host environment it must encounter during infection. This adaptability largely relies on signal transducing systems which turn on complex transcription networks [1]. Bacterial two component systems (TCSs) are coupled signal sensing and transduction apparatus which comprise of two proteins, a transmembrane sensor kinase (SK) responsible for sensing signal(s) and a cognate cytoplasmic response regulator (RR), that translate environmental information into a physiological response [2,3]. Underlying the functional diversity of TCSs is a common set of chemical reactions by which the activated RR impacts expression of numerous genes in response to signal(s), that is first sensed by a SK which then activates the cognate regulator via a sequential phosphotransfer reaction [4]. *M. tuberculosis* genome encodes for 12 complete TCSs, fewer relative to other bacterial species with comparable genome size [5–7].

Numerous studies have uncovered that several TCSs, namely DosRST, PhoPR, MprAB, SenX3-RegX3, PdtaRS and MtrAB, have defined roles to *in vivo* virulence, as determined by various infection models utilizing immune -compromised and competent mice, guinea pigs, rabbits and non-human primates [8–15]. Of these, the PhoP/PhoR pair of proteins of *M. tuberculosis* form a specific TCS that functions with a key regulatory role in controlling cell-wall lipid composition and virulence [16–19], immediate and enduring hypoxic responses, aerobic and anaerobic respiration [20,21], secretion of major virulence factors [22,23], stress response and persistence [20,21,24,25] [for reviews see [26], [27]]. Notably, PhoP controls synthesis of three classes of polyketide-derived acyltrehaloses, namely sulfolipids, di-acyl trehaloses (DATs) and poly acyltrehaloses (PATs) [13,18]. In keeping with this, avirulent *M. tuberculosis* H37Ra with a genomic copy of PhoP harbouring a single nucleotide polymorphism (SNP) within its effector domain, shows growth attenuation [22,28] and lacks polyketide-derived acyltrehaloses relative to virulent *M. tuberculosis* H37Rv [29]. More importantly, virulence and PAT/DAT biosynthesis is restored in *M. tuberculosis* H37Ra expressing copy of *M. tuberculosis* H37Rv *phoP* gene [22,30]. Transcriptomic study reveals that ~2% of the *M. tuberculosis* H37Rv genome is regulated by PhoP at the transcriptional level [13,31]. Therefore, with inactivation of *phoPR* the tubercle bacilli demonstrate a strikingly reduced multiplication in macrophages, and consequently the *phoPR* deletion strain is considered in vaccine trials [32].

Studies aimed to understand functioning of the transcription factor uncovered that *ΔphoP*-H37Rv complemented with the phosphorylation defective PhoP fails to rescue complex lipid

biosynthesis [30], suggesting P~PhoP dependent regulation of mycobacterial genes [30]. However, we still do not fully understand the mechanism of activation of the virulence-associated PhoP/PhoR system. Although a great deal is known about the mechanism of functioning of PhoP, relatively much less is known about PhoR, the HK sensor of this complex. PhoR is a homodimer, and each subunit consists of two transmembrane helices which flank an N-terminal sensor domain (extra-cytosolic), a HAMP domain, a DHp (dimerization and phospho-transfer) domain, and a cytoplasmic CA (catalytic and ATP-binding) domain, respectively. The DHp domain plays a central role of the HK sensor function. These include interactions with the ATP binding domain and autophosphorylation at His 259, contacting the partner RR to transfer phosphate group (for the latter's activation), and accepting signal(s) from the upstream sensor domain to regulate phosphorylation [33]. While *in vitro* studies have shown autophosphorylation of PhoR and phospho-transfer to PhoP [34], *in vivo* phosphorylation of PhoR and PhoP is yet to be demonstrated. More importantly, the signal(s) that activates PhoP/PhoR TCS remains unknown.

In this study, we show that PhoR is a direct sensor of acidic pH conditions, which subsequently activates PhoP via phosphorylation. Using mycobacterial protein fragment complementation (M-PFC) -based screening coupled with phosphorylation assays, we identify PrrB as a bona fide phospho-donor of PhoP. We also demonstrate that in addition to its kinase activity, PhoR functions as a phosphatase of P~PhoP, and this activity of PhoR connects signal-dependent pool of intra-mycobacterial P~PhoP with downstream functioning of the regulator. Using mutant PhoR proteins, we provide evidence to show that contrasting kinase and phosphatase functions of the PhoR remain critically important to determine the final output of the acidic pH-inducible PhoP regulon. Thus, PhoR functions remain critical for its central role in mycobacterial adaptation to low pH conditions within the host macrophage phagosome. In keeping with these results, a PhoR-deleted *M. tuberculosis* remains significantly attenuated in macrophages and animal models.

## Results

### Acidic pH and high salt conditions of growth promotes phosphorylation of PhoR and PhoP

Because SK of a TCS is often responsible for sensing environmental signal(s), abundance of the SK transcripts does not necessarily correlate with activation of a TCS. However, determining the amount of phosphorylated SK/RR represents a more direct measure of activation of the system [35]. Thus, to examine whether PhoR can sense activation signal(s) during *in vitro* growth conditions, we studied phosphorylation of PhoR using phos-tag gels [36]. In this assay, phos-tag gels effectively resolve phosphorylated and unphosphorylated forms of the SK/RR in cell lysates of bacteria grown under specific environmental conditions. These experiments were carried out in WT-H37Rv background in which His-tagged PhoR was expressed ectopically from the promoter of the 19-kDa antigen of p19KPro [37], and the amount of P~PhoR was detected by Western blotting using antibodies that recognize His-tagged PhoR (Fig 1A). When WT bacilli was grown under acidic (pH 4.5) conditions, >90% of the total PhoR protein was in the phosphorylated (P~PhoR) form compared to ~5% of P~PhoR (compare lane 1 and lane 2) for cells grown under normal (pH 7.0) conditions. In sharp contrast, >90% of PhoR was available in the unphosphorylated form in mycobacterial cells grown in presence of 5 mM diamide (under oxidative stress) (lane 3). Further, P~PhoR accounted for ~90% of total PhoR in extracts of bacterial cells grown in presence of 250 mM NaCl (salt stress) (lane 4). From these results, we conclude that *M. tuberculosis* PhoR is capable of sensing low pH and high salt

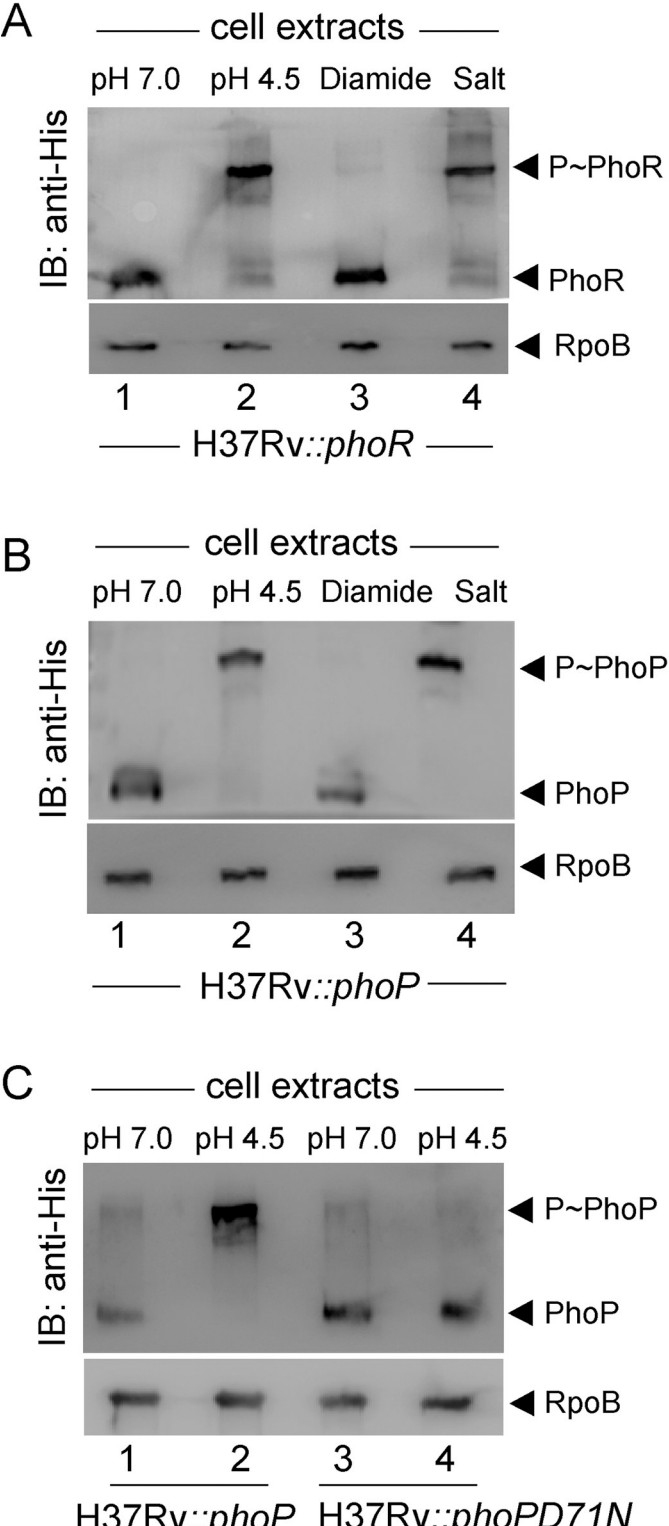

**Fig 1. Mycobacterial growth under conditions of acidic pH and high salt concentration promote phosphorylation of PhoR and PhoP.** Phos-tag analysis of cell lysates of WT-H37Rv harbouring His-tagged (A) PhoR, (B) PhoP or (C) phosphorylation-defective PhoPD71N proteins, were grown either under normal conditions (pH 7.0) or under indicated stress conditions, as described in the Methods. PhoR and PhoP were detected by Western blotting using anti-His antibodies, and RpoB (as a loading control) was detected in cell lysates using anti-RpoB antibody (Abcam). Data are representative of two independent experiments.

conditions of growth. Notably, these results are consistent with a previous study suggesting low pH and chloride as synergistic cues that intracellular *M. tuberculosis* responds to [38].

In order to probe phosphorylation of the cognate RR PhoP, we next expressed full-length *M. tuberculosis* PhoP protein in WT-H37Rv as described above. Cells were grown under stress conditions and presence of P~PhoP was probed in cell extracts using anti-His antibody (Fig 1B). Remarkably, we could detect majority of P~PhoP (>80%) in extracts of mycobacterial cells grown under low pH (lane 2) or high salt (lane 4) conditions. In contrast, for cells grown under normal conditions (lane 1) or oxidative stress (lane 3), majority of the PhoP protein was in the unphosphorylated state. Thus, we conclude that stress-specific *in vivo* phosphorylation of PhoP is most likely dependent on phosphorylation of the cognate kinase, PhoR. As a control, we repeated an identical experiment using WT-H37Rv ectopically expressing His-tagged PhoPD71N, a phosphorylation defective mutant of PhoP (Fig 1C). Although mycobacterial cells expressing WT-PhoP grown at pH 4.5 showed a significantly higher abundance of P~PhoP compared to cells grown at pH 7.0 (compare lane 1 and lane 2), we could not detect P~PhoP from mycobacterial cells expressing PhoPD71N (compare lane 3 and lane 4) under identical conditions. It should be noted that the weak signal of P~PhoP in the Western blot for mycobacterial cells grown at pH 7.0 remains irreproducible. From these results, we conclude that low pH or high salt concentration promotes *in vivo* phosphorylation of *M. tuberculosis* PhoP.

## PhoP phosphorylation under acidic pH or high salt conditions requires the presence of PhoR

To examine whether PhoR is necessary for acidic pH and high salt concentration dependent phosphorylation of PhoP. We next utilized 'mycobacterial recombineering' [39] to construct a PhoR-deleted *M. tuberculosis* H37Rv (*ΔphoR*-H37Rv). The mutant construction is schematically shown in Fig 2A, and described in the Methods. To verify the construct, PCR amplification was carried out using a pair of *phoR*-specific internal primers (FPphoRInt / RPphoRInt) and genomic DNA of *ΔphoR*-H37Rv (S1A Fig). The results showed that *phoR*-specific amplicon was absent in the mutant relative to the WT-H37Rv (compare lane1 and lane 2). However, the complemented mutant utilizing an integrative plasmid harbouring a copy of *phoR* gene showed the presence of *phoR*-specific amplicon (lane 3). As controls, genomic DNA of these three strains yielded full-length *phoP*-specific amplicon (lanes 5–7). Also, *ΔphoR*-H37Rv was confirmed by Southern hybridization experiment (S1B Fig). Although a *hrcA*-specific amplicon (~9.6 -kb) could be detected in both WT and mutant strain using gene-specific end-labelled primers, *phoR*-specific (~2 -kb) amplicon was detectable only in case of WT-H37Rv (compare lane 2 and lane 3). Consistent with these results, RT-qPCR experiments show that mRNA level of *phoR* remains undetectable in the mutant relative to WT-H37Rv, and expression of PhoR is significantly restored in the complemented mutant (Fig 2B). The oligonucleotides used in cloning/amplifications, and the plasmid constructs used in expression of fusion proteins are listed in S1 and S2 Tables, respectively. Next, *in vivo* phosphorylation assays using *ΔphoR*-H37Rv under varying conditions of growth showed that P~PhoP was undetectable in low pH or high salt conditions of growth (lanes 1–4, Fig 2C), suggesting significantly reduced phosphorylation of PhoP in a PhoR-deleted strain, However, under identical conditions we could detect effective phosphorylation of PhoP in WT-H37Rv grown under low pH conditions (compare lanes 5 and 6). From these results, we conclude that abundance of PhoP~P during mycobacterial growth under acidic pH depends on the presence of the cognate kinase PhoR.

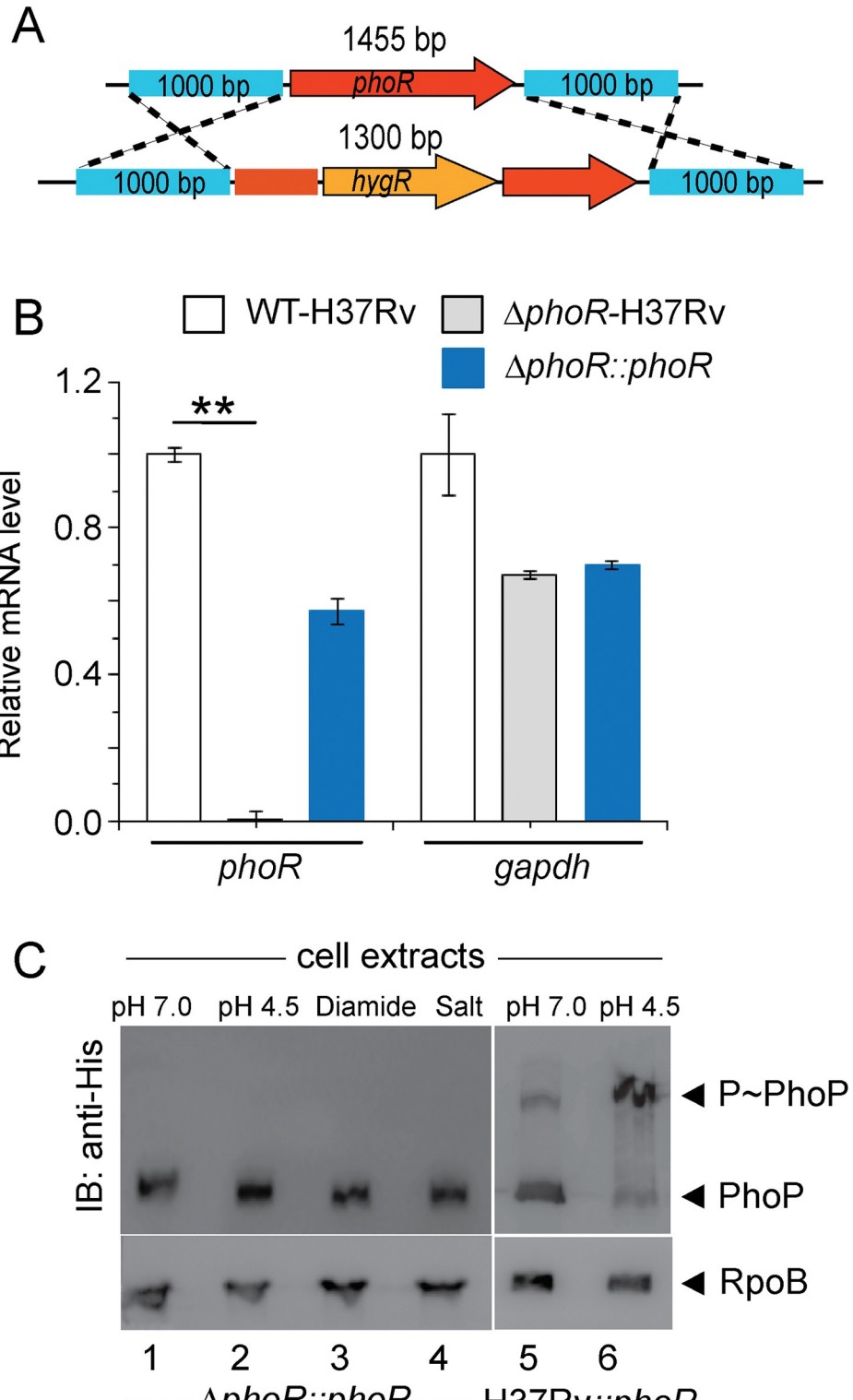

**Fig 2. Acidic pH/high salt concentration-dependent phosphorylation of PhoP requires the presence of PhoR.** (A) Schematic diagram showing construction of PhoR-deleted *M. tuberculosis* H37Rv (*ΔphoR*-H37Rv) by 'mycobacterial recombineering'. (B) We utilized RT-qPCR experiments to compare mRNA level of *phoR* in the mutant and WT-H37Rv, as described in the Methods. Importantly, expression of *phoR* remains undetectable in the mutant relative to WT-H37Rv, but is significantly restored in the complemented mutant. As a control, *gapdh* levels remain

comparable in WT-H37Rv, *ΔphoR*-H37Rv mutant and the complemented mutant strain. The results are derived from average of biological duplicates, each with one technical repeat (**P≤0.01). (C) To probe *in vivo* phosphorylation, Phos-tag Western blot analysis were performed using cell lysates prepared from *ΔphoR*-H37Rv, and WT-H37Rv harbouring His-tagged PhoP protein, respectively. The relevant strains were grown under normal or indicated stress conditions, as described in the Methods. Sample detection and data analysis were carried out as mentioned in the legend to Fig 1.

## PhoR impacts global expression of the PhoP regulon

Numerous studies have shown that *M. tuberculosis* growth and transcriptional control are tightly regulated by environmental pH [40–42]. Along the line, global transcriptional response to acid stress suggests possible involvement of multiple regulators including *phoPR* in response to the cue of phagosomal acidification. [13,18,43,44]. To investigate regulatory influence of PhoR on mycobacterial gene expression, we compared transcriptomes of WT- and *ΔphoR*-H37Rv, grown under normal (pH 7.0) and acidic (pH 4.5) conditions of growth. The RNA-sequencing results (S3 and S4 Tables) demonstrate that ~167 genes of WT-H37Rv are induced under low pH (pH 4.5) relative to mycobacterial cells grown under normal pH (pH 7.0) (S2A–S2B Fig and also see S4 Table). However, under identical conditions, *ΔphoR*-H37Rv displayed a low pH-inducible expression of only ~ 31 genes (Fig 3A), suggesting that expression of ~81% of acidic pH-inducible genes require mycobacterial *phoR* locus (see S4 Table). Further, ~22 genes displayed a low pH-dependent down-regulation in *ΔphoR*-H37Rv relative to normal pH (S2C Fig). Note that these results are organized based on genes annotated in the *M. tuberculosis* H37Rv genome using transcriptomic data reported in this study and previously-reported high throughput data of the PhoP regulon [31]. S2D–S2E Fig show the Volcano plots of RNA-seq data assessing pH inducible expression of genes for WT-H37Rv and *ΔphoR*-H37Rv, respectively. Importantly, majority of the PhoR- controlled genes including genes from diverse functional categories like regulatory proteins, components of information pathways, virulence, detoxification and adaptation, metabolism and respiration, cell wall biosynthesis, lipid metabolism, and PE/PPE proteins etc., were part of the PhoP regulon. Along the line, a comparison of transcriptomic data of *phoR*-mutant (reported in this study) and a previously reported *phoP*-mutant, grown under normal conditions (pH 7.0; see Methods) [31] appears to suggest regulation of 157 and 152 genes (p ≤ 0.05) by PhoR and PhoP, respectively, displaying an overlap of ~42 genes.

Next, we measured mRNA levels of representative low pH-inducible genes in *ΔphoR*-H37Rv relative to WT-H37Rv (Fig 3B). In agreement with the RNA-seq data, RT-qPCR experiments demonstrate a significant down-regulation of representative PhoP regulon genes in *ΔphoR*-H37Rv. The fact that PhoR functions as an activator of these genes was further confirmed as stable expression of *phoR* in the complemented mutant could largely restore low pH-inducible gene expression. We previously showed that *phoP* expression level is reproducibly higher in the complemented mutant (relative to the WT bacilli) both under normal conditions as well as during growth under low pH conditions [45]. These results possibly account for elevated mRNA levels of a few representative genes in the complemented mutant relative to WT-H37Rv. To assess whether presence of the *phoR* locus impacts on DNA binding activity of PhoP, we compared recruitment of the regulator *in vivo* within its promoters in WT-H37Rv and *ΔphoR*- H37Rv by chromatin immunoprecipitation (ChIP) assay (Fig 3C). While qPCR measurements showed effective recruitment of PhoP in WT-H37Rv, under identical conditions a significantly weaker PhoP recruitment to its target sites was apparent in *ΔphoR*-H37Rv. Thus, we conclude that *in vivo* recruitment of PhoP requires the cognate kinase, PhoR. These results are consistent with previously- reported findings suggesting (a) PhoR as an obligate SK of PhoP [33] and (b) importance of phosphorylation of PhoP in high-affinity DNA binding

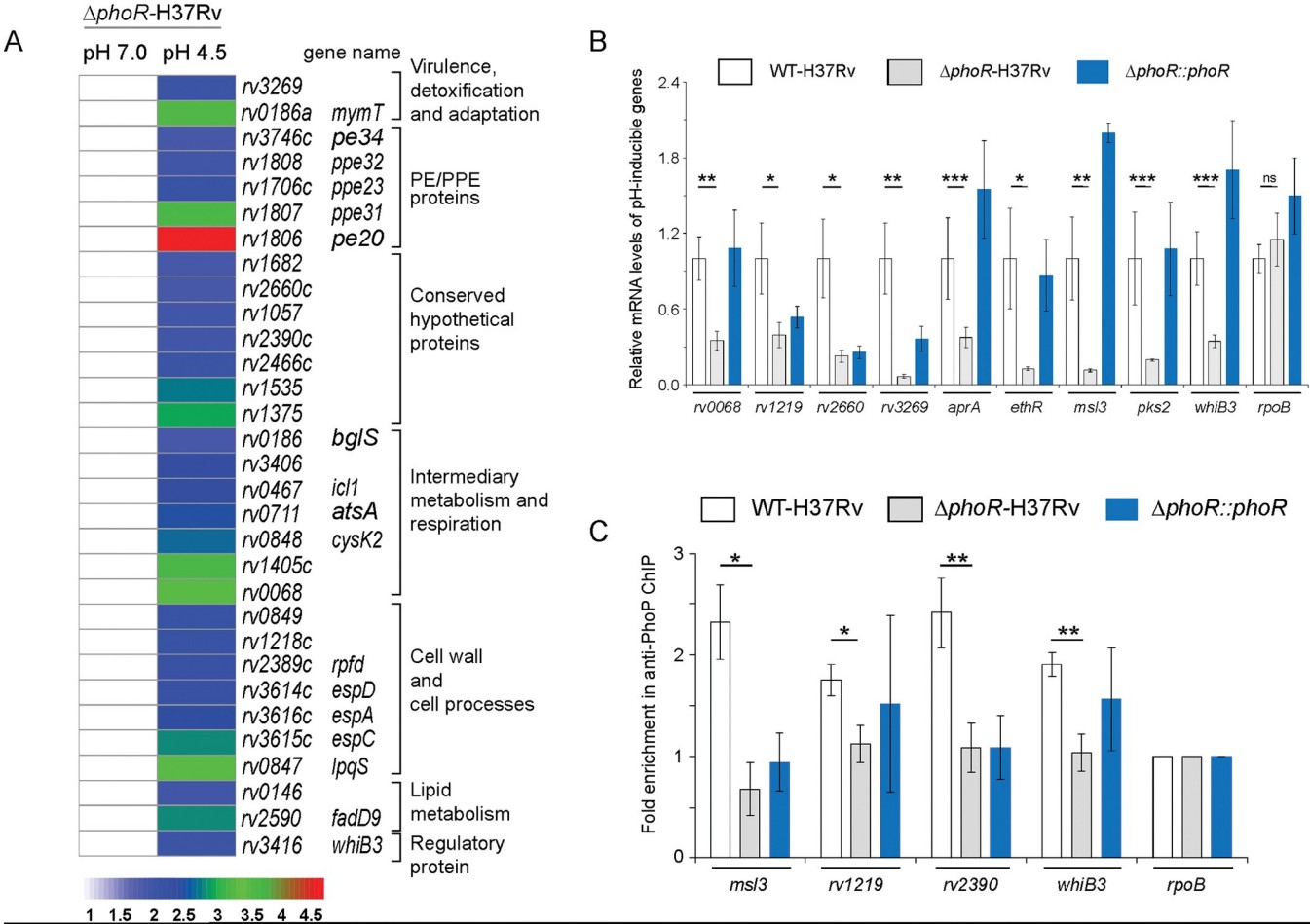

**Fig 3. Global expression of acid-inducible mycobacterial genes require the *phoR* locus.** (A) RNA-sequencing derived heat-map showing ~ 31 low pH-inducible genes that are differentially regulated in *ΔphoR*-H37Rv (>2.0 fold; p< 0.05) grown under low pH (pH 4.5) conditions compared to normal conditions (pH 7.0) of growth. The data, which represent average of two biological replicates, list significantly activated pH-inducible genes, majority of which are part of the PhoP regulon. (B) PhoR-dependent expression of acidic pH-inducible genes were examined by comparing expression of representative mycobacterial genes in WT-H37Rv, *ΔphoR*-H37Rv, and complemented strain, grown under normal and acidic pH conditions. The data show plots of average values from biological duplicates, each performed with one technical repeat (*P≤0.05; **P≤0.01; ***P≤0.001). (C) To examine *in vivo* recruitment of PhoP within its target promoters, ChIP was carried out using anti-PhoP antibody followed by qPCR using IP samples from WT-H37Rv and *ΔphoR*-H37Rv (see Methods section for further details). To assess fold enrichment, each data point was compared with the corresponding IP sample without adding antibody. The experiments were carried out as biological duplicates, each with at least one technical repeat (*P≤0.05; **P≤0.01). Non-significant differences are not indicated.

and transcription regulation [30]. The sequences of oligonucleotides utilized in RT-qPCR and ChIP experiments are listed in S5 Table.

To verify PhoR-independent expression of acidic pH inducible genes, we next compared expression of a few representative genes in *ΔphoR*-H37Rv, grown under acidic and normal conditions of growth. Our results uncover that consistent with the RNA sequencing data (see Fig 3A), expression of these genes display *phoR* independent acidic pH-inducible expression (S3A Fig). We also studied expression of these genes in WT-H37Rv (S3B Fig) and the complemented *ΔphoR*-H37Rv (S3C Fig) and observed similar pH-inducible expression profile as that of *ΔphoR*-H37Rv. Further, in keeping with these results, ChIP-qPCR experiments demonstrate *phoR*-independent acidic pH-inducible PhoP recruitment within the target promoters (S3D Fig). Together, these results establish a *phoR*-independent mechanism of pH-inducible expression of a few mycobacterial genes.

## Probing dual functioning of sensor kinase PhoR

X-ray structure of *M. tuberculosis* PrrA, the closest family member of PhoP reveals that phosphorylation of the regulator appears to impact DNA binding and transcriptional control by the regulator [46]. Consistent with this, phosphorylation of PhoP strikingly impacts mycobacterial cell shape and morphology via its major regulatory control of genes responsible for complex lipid biosynthesis [30]. However, till date, it remains to be determined how cytoplasmic pool of phospho-PhoP is regulated. In the absence of an inducing stimulus, dephosphorylation of RR is critically important to ensure the TCS pathway gets reset. To examine the possibility that PhoR can function as a phosphatase, we designed an *in vitro* assay as shown schematically in Fig 4A. In this experiment, PhoP was phosphorylated by radio-labelled ATP and acetyl-kinase as described elsewhere [47], and incubated with equimolar PhoRC [comprising C-terminal 293 residues of PhoR [34]] for varying lengths of time (Fig 4B–4E). Interestingly, we observed a time-dependent dephosphorylation of P~PhoP in presence of PhoRC (Fig 4B). However, heat-denatured PhoRC, under identical conditions failed to dephosphorylate P~PhoP (Fig 4C). As a control, recombinant PrrBC (see Methods) failed to dephosphorylate P~PhoP (Fig 4D), suggesting PhoR-specific phosphatase activity. As an additional control, PhoRC was incubated with radio-labelled P~DosR under identical conditions (Fig 4E). The results unambiguously demonstrate that PhoRC is incapable of dephosphorylating P~DosR. From these data, we surmise that PhoR functions as a specific phosphatase to maintain intra-mycobacterial cellular pool of P~PhoP.

To determine which residue(s) of the DHp domain of PhoRC contribute to phosphatase activity, we next aligned α1-helix of the DHp domain of PhoR sequence with its family members, known to have exhibited phosphatase activity (Fig 4F). A closer inspection of the sequence comparison reveals the presence of a conserved EXXT/N motif adjacent to His-259, the primary phosphorylation site of PhoR [34]. This is consistent with a few orthologues such as, EnvZ [48] and CPXA [49], which exhibit cognate RR-specific phosphatase activity attributable to either of the sequence motifs DXXXQ or E/DXXT/N, present within the α1 helix of DHp domain [47,50]. S4A Fig shows the structural model of PhoRC indicating the most conserved residues of α1 helix.

To examine role of the EXXT/N motif (residues 260–263) of PhoR in phosphatase activity, we replaced the most conserved residue (E260) by site-directed mutagenesis, and constructed a point mutant, PhoRCE260D. As a control, we also constructed PhoRCD282E, a mutant PhoRC replacing an aspartate with a glutamate further downstream of the α1 helix. The WT- and mutant PhoRC proteins were purified [34] (see S4B Fig), and *in vitro* autophosphorylation experiments using recombinant proteins demonstrate that except PhoRCH259Q (lane 1, S4C Fig), the other two mutants (lanes 2–3) undergo auto-phosphorylation as that of WT PhoRC (lane 4). This is consistent with His-259 as the primary site of phosphorylation of PhoR [34]. Also, these results further suggest that the point mutants are structurally stable and functionally active.

Next, phosphatase assays were carried out as described in Fig 4B–4E using equimolar WT- or mutant PhoRC (recombinant) proteins (Fig 4G–4I). As expected, phosphorylation-defective PhoRCH259Q was unable to function as a phosphatase (lanes 1–5, Fig 4G), whereas WT-PhoRC showed efficient phosphatase activity under identical conditions (lanes 6–10, Fig 4G). Interestingly, PhoRCE260D was found to be remarkably ineffective for phosphatase activity relative to WT-PhoRC (compare lanes 1–5, and lanes 6–10, Fig 4H). In contrast, PhoRCD282E, under identical conditions examined, displayed a comparable phosphatase activity as that of WT-PhoRC (compare lanes 1–5, with lanes 6–10, Fig 4I). Thus, we conclude that although H259 remains the primary phosphorylation site, the adjacent conserved residue

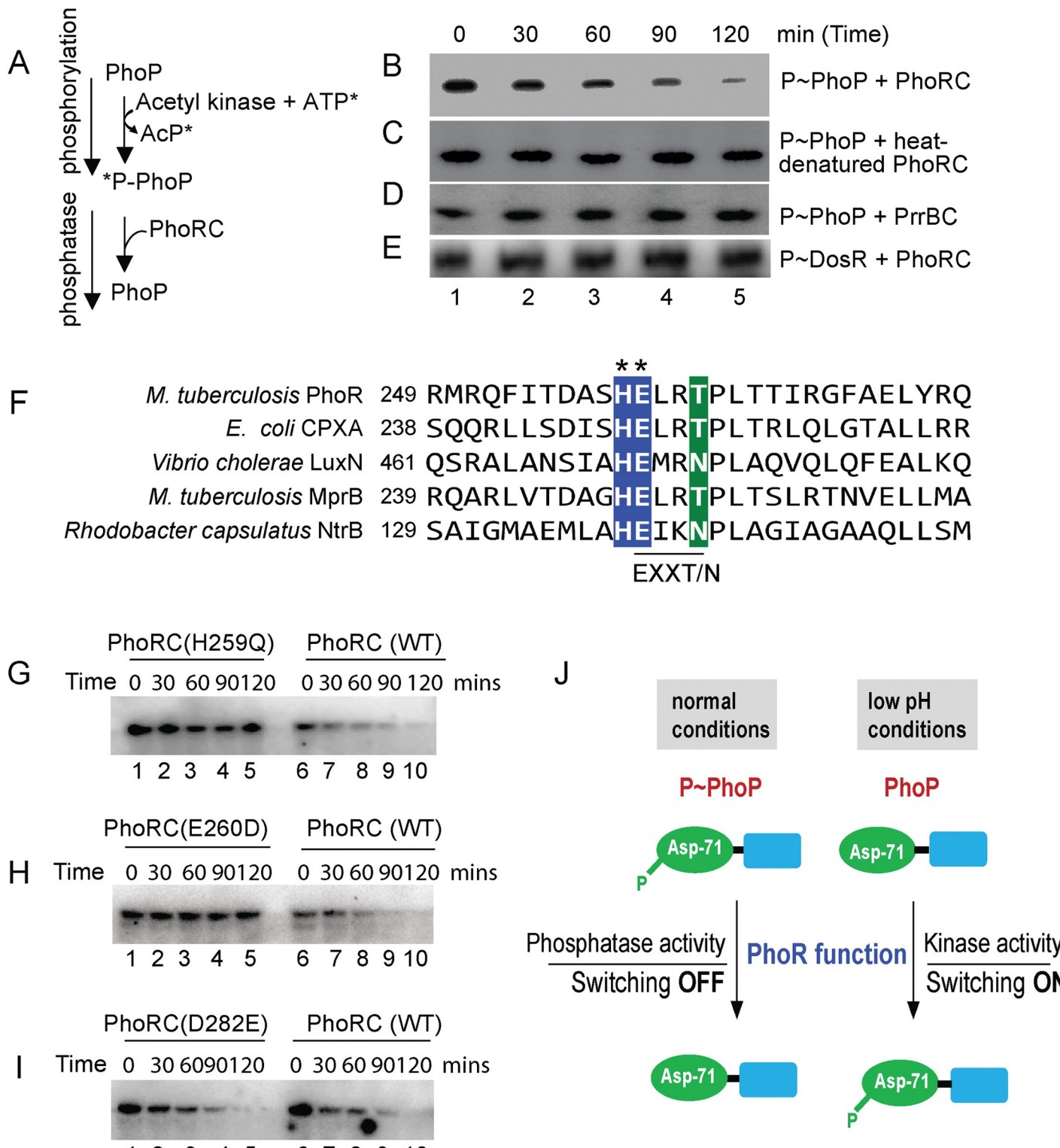

**Fig 4. Identifying motif/residues responsible for phosphatase function of PhoR.** (A) Scheme showing phosphorylation of PhoP by $\gamma$-$^{32}$P-ATP and acetyl kinase and dephosphorylation of P-PhoP. (B-E) Time-dependent dephosphorylation of P-PhoP was examined by adding recombinant PhoRC (B), heat-denatured PhoRC (C), and purified PrrBC (D) as described in the Methods. (E) To examine specificity of newly-identified PhoR phosphatase activity, P~DosR was used as a substrate to examine possible dephosphorylation by adding purified PhoRC. The reactions were followed by SDS-PAGE analyses and digited by a phosphorimager. Each data is representative of at least two independent experiments. (F) DHp domain of PhoR was aligned with other SKs, which displayed phosphatase activity. The most

conserved residues of the EXXT/N motif are indicated by asterisks. (G-I) To investigate the effect of mutations on the phosphatase function of PhoRC, purified WT and mutant proteins were incubated with P~PhoP for indicated times as described in Fig 4B. In all cases, reactions were followed by SDS-PAGE analyses, and digitized by phosphorimager. (J) Schematic model showing balancing act of dual functioning of PhoR as a kinase (activation) and phosphatase (repression). Mutant PhoR protein, defective for kinase activity, fails to phosphorylate PhoP and thereby impact acid-inducible expression of the PhoP regulon. In contrast, phosphatase activity of PhoR dephosphorylates P~PhoP either to reverse active mycobacterial PhoP regulon in the absence of an inducing signal or to prevent unnecessary 'triggering on' of PhoP regulon. In summary, PhoR regulates net phosphorylation of PhoP by its dual functioning (kinase/phosphatase) to control context-sensitive expression of the PhoP regulon.

E260 of PhoR is critically required for phosphatase activity of the SK. Together, these results suggest importance of a limited number of residues of the DHp α1-helix in kinase/phosphatase dual functioning of PhoR. Based on these results, we present a schematic model (Fig 4J) of pH-dependent mycobacterial adaptive program via phosphorylation-coupled 'switching on' and dephosphorylation-mediated 'switching off' of the PhoP regulon utilizing contrasting kinase and phosphatase activities of PhoR, respectively.

## Screening for PhoP-interacting mycobacterial SK(s)

Based on activation of a set of low pH -inducible PhoP regulon genes in *ΔphoR*- H37Rv, we speculated non-canonical interactions and possible phospho-transfer between non-cognate SK(s) and PhoP. Cross-talk between TCS proteins has been considered a potential mechanism for complex stimulus integration and signal transduction in bacteria because of considerable high sequence and structural similarity in members of SK and RR family of proteins [51–53]. To probe SK-PhoP interaction(s), we utilized mycobacterial protein fragment complementation (M-PFC) using *M. smegmatis* as the surrogate host [54] (Fig 5A and 5B). According to this assay, two interacting mycobacterial proteins which are expressed as C-terminal fusions with complementary fragments of mDHFR (murine dihydrofolate reductase), provides bacterial resistance to trimethoprim (TRIM) because of functional reconstitution of the full-length enzyme. *M. smegmatis* harbouring the corresponding plasmids were grown on 7H10/Kan/Hyg in the presence or absence of 15 μg/ml TRIM. Strikingly, *M. smegmatis* cells expressing both PrrB/PhoP proteins displayed clear growth in presence of TRIM, whereas none of the other *M. tuberculosis* SKs under identical conditions, displayed *in vivo* interaction with PhoP. Although we cannot rule out the possibility of other kinases with weaker affinities for PhoP remain undetectable in this assay, collectively these results, under the conditions examined, suggest specific protein-protein interaction between PrrB and PhoP. Importantly, *M. smegmatis* harbouring vectors lacking an insert (empty) did not grow on 7H10/TRIM plates, while all other *M. smegmatis* strains displayed normal growth on 7H10 plates lacking TRIM. S6 and S7 Tables list the sequences of oligonucleotides, and relevant plasmids, respectively, used in cloning for M-PFC assays.

We also verified PrrB-PhoP interaction *in vitro* by pull-down assays using GST-PhoP and His₆-tagged PrrBC (cytoplasmic region of PrrB comprising C-terminal residues 200 to 446) (Fig 5C), cloned, expressed and purified as described in the Methods. In this experiment, GST-tagged PhoP was bound to glutathione-Sepharose, and then incubated with purified PrrBC. Upon elution of column-bound proteins, both proteins were detected in the same fraction (lane 1). However, a detectable signal was absent in case of only GST-tag (lane 2) or only resin (lane 3), further suggesting that PhoP interacts with PrrB. Also, we have carried out a pull-down experiment using GST-tagged PhoRC and His-tagged PrrA (Fig 5D). Here, we were unable to detect presence of His-PrrA and GST-PhoRC in the same fraction (lane 1) although GST-PhoRC and His-PhoP coeluted in the same fraction (lane 2). From these results we conclude that although PhoR does not appear to interact with PrrA, *M. tuberculosis* PrrB interacts with PhoP.

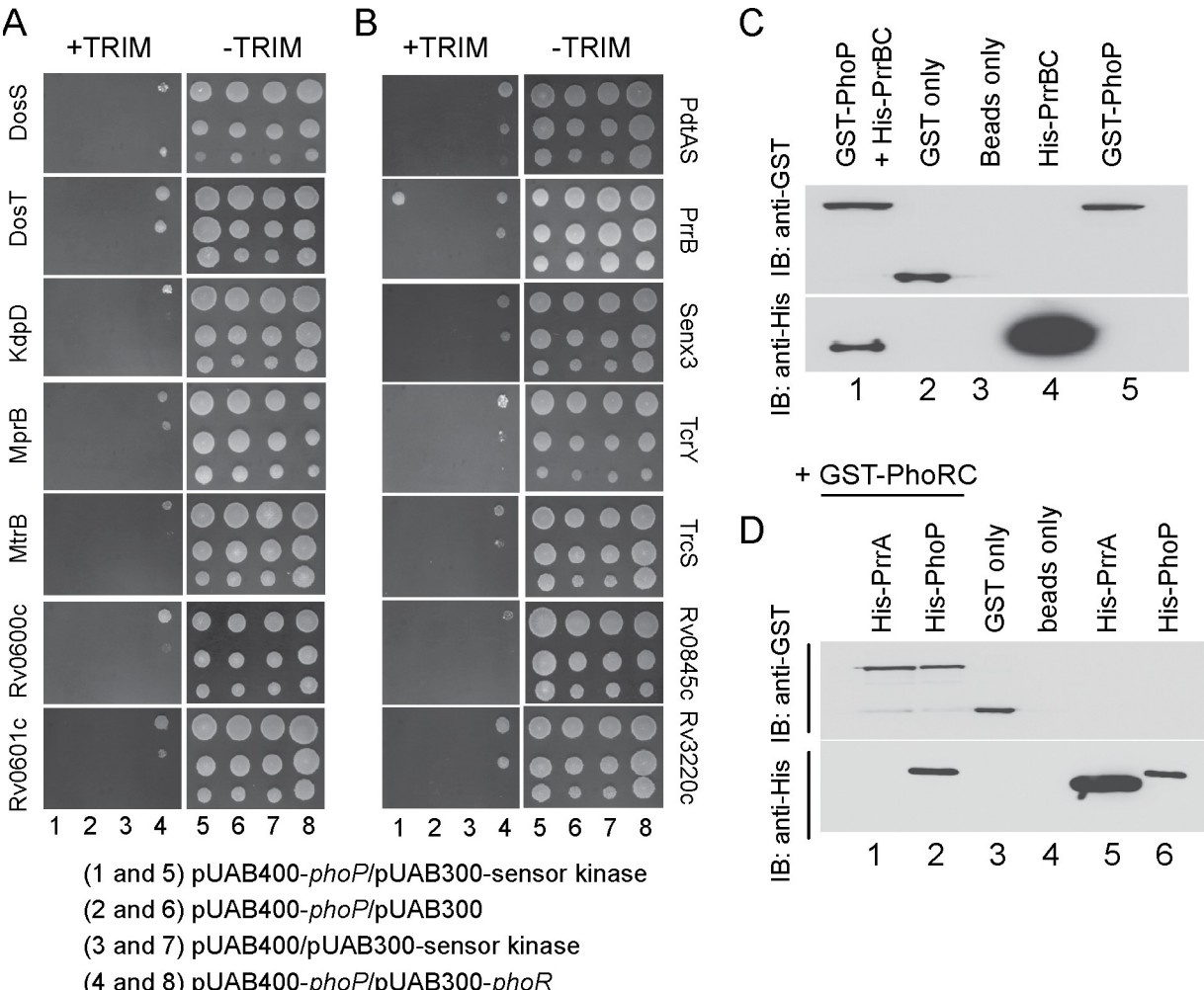

**Fig 5. Screening to probe PhoP interacting mycobacterial SK(s).** (A-B) M-PFC experiments co-expressing *M. tuberculosis* PhoP and SKs were used as a screen using *M. smegmatis* as the surrogate host. Co-expression of pUAB400-*phoP*/pUAB300-SK (indicated) pair (columns 1 and 5) relative to empty vector controls, pUAB400-*phoP*/pUAb300 (columns 2 and 6), or pUAB400/pUAB300-SK (columns 3 and 7), supporting *M. smegmatis* growth in presence of TRIM is suggestive of specific interaction. Note that in each case the three spots moving downwards in each panel represent spotting of cells at three different dilutions, namely undiluted, 10-fold and 100-fold dilutions, respectively. While co-expression of pUAB400-*phoP*/pAUB300-*phoR* (columns 4 and 8) in *M. smegmatis* displaying growth in presence of TRIM served as a positive control, growth of all the strains in absence of TRIM (columns 5–8) validated the assay. (C-D) To examine interactions *in vitro*, purified His$_6$-PrrBC or purified His$_6$-PrrA and His$_6$-PhoP were incubated with glutathione-Sepharose that was bound to GST-PhoP (C) or GST-PhoRC (D), respectively. Bound proteins (lane 1, panel C or lanes 1–2, panel D) were detected by Western blot using anti-GST (upper panel) or anti-His (lower panel) antibodies. Replicate experiments used glutathione Sepharose bound to GST (lane 2, panel C and lane 3, panel D), or the resin alone (lane 3, panel C or lane 4, panel D); lanes 4 and 5, panel C resolve His-PrrBC and GST-PhoP, respectively; lanes 5, and 6, panel D resolve His-PrrA, and His-PhoP, respectively.

## Phosphorylation of PhoP by a non-cognate SK

Having shown a specific interaction between PrrB and PhoP by M-PFC and *in vitro* pull-down assays, we investigated possible phosphotransfer between the two non-cognate partners. To facilitate purification, full-length PhoP and the cytoplasmic portion of PhoR (PhoRC) was over-expressed as His$_6$-tagged fusion proteins, as described [34]). An N-terminally truncated form of PhoR (PhoRC, residues 193 to 485) and PrrB (PrrBC, residues 200 to 446) lacking the membrane spanning region were chosen, as over-expression of full-length sensor kinase genes is often toxic [55]. Additionally, PhoRC, and PrrBC were preferred to avoid insolubility due to

the presence of hydrophobic amino acid stretches. PrrBC was constructed based on a sequence alignment result with members of the SK family of proteins. Note that full length PhoP (247 amino acids) is comparable to the cytoplasmic region of PrrB protein (PrrBC comprising 247 amino acids) in size, justifying use of the N-terminal domain of PhoP (PhoPN, 141 amino acids), which was previously shown to function as the phosphoacceptor domain as effectively as the full-length PhoP [56]. Auto-phosphorylation of PhoRC/PrrBC and subsequent phospho-transfer to PhoPN/PrrA were performed as described previously [34,56] and are schematically shown in Fig 6A. For autophosphorylation, ~ 2.5 μM PhoRC/PrrBC was incubated in a phosphorylation mix containing ~25 μM γ$^{32}$P-ATP. After incubation of 30 minutes at 25˚C, ~ 5 μM purified PhoPN or PrrA was added to the phosphorylation mix, samples were incubated for indicated times, aliquots withdrawn, and the products analysed by SDS-PAGE. Importantly, recombinant P~PrrBC could effectively phosphorylate PhoPN (lanes 2–5, Fig 6B). However, under identical conditions P~PhoRC was unable to phosphorylate recombinant PrrA (lanes 7–10, Fig 6B). While this observation is consistent with lack of detectable interaction between PhoRC and PrrA (Fig 5D), phosphotransfer between labelled PhoRC and PhoPN was shown as a control (lanes 12–14, Fig 6B). Taken together, these results suggest specificity of PrrB -dependent phosphorylation of PhoP.

Next, to further probe activation of PhoP by specific SKs, we adopted a CRISPRi- based gene knock-down approach [57]. The *phoR* and *prrB* knock-down constructs are described in the Methods, and relative mRNA levels compared expression of genes of interest in the corresponding knock-down strains relative to WT-H37Rv (S5A Fig). Note that both the *phoR*-KD and *prrB*-KD mutants displayed significantly down-regulated expression of the corresponding target genes relative to WT bacilli. However, as controls *prrB* expression in *phoR*-KD and *phoR* expression in *prrB*-KD showed insignificantly higher expression relative to the WT-bacteria. Next, we grew these strains under low pH conditions to examine expression of a few representative genes, which showed either *phoR* -dependent or *phoR*-independent pH-inducible activation (S3–S4 Tables). Importantly, representative genes displaying *phoR*-dependent pH-inducible expression (see Fig 3B) clearly showed a significantly lowered expression in *phoR*-KD strain relative to WT-H37Rv under acidic pH (Fig 6C). However, consistent with the results reported in RNA-seq data (Fig 3A), *phoR*-KD mutant failed to impact expression of the genes which show *phoR*-independent pH- inducible expression. S5B Fig shows RNA-seq derived heat-map of these genes from WT- and *ΔphoR*-H37Rv, under low pH conditions of growth. More importantly, genes which remained unaffected in *phoR*-KD strain, showed a significant inhibition of expression in *prrB*-KD strain relative to the WT bacilli under identical conditions (Fig 6D). In contrast, expression of genes showing significant inhibition in *phoR*-KD mutant, remained unaffected in *prrB*-KD, under identical conditions examined. Collectively, these data suggest that although PhoR accounts for activation of PhoP regulon under acidic pH, PhoP activation is partly attributable to mycobacterial PrrB. These results are consistent with PrrB -dependent phosphorylation of PhoP, and for the first time, connect two SKs to integrate signal-dependent activation of PhoP, perhaps uncovering an effective means of mycobacterial adaptability to varying intracellular environments.

In an attempt to investigate under which condition PrrB is phosphorylated, we next grew WT-H37Rv expressing His-tagged PrrB under nitrogen limiting conditions and acidic pH, respectively, and assessed *in vivo* phosphorylation of the sensor kinases by phos-tag analysis (Fig 6E). We included nitrogen limiting conditions, because mycobacterial PrrAB was reported to be induced under nitrogen limiting conditions [58]. Also, a recent study uncovered that PrrA as a transcription factor modulates mycobacterial stress response during low pH conditions [59]. Expectedly, PrrB was mostly in the unphosphorylated form under normal conditions (lane 1) of growth. Also, we failed to observe phosphorylation of PrrB in cells

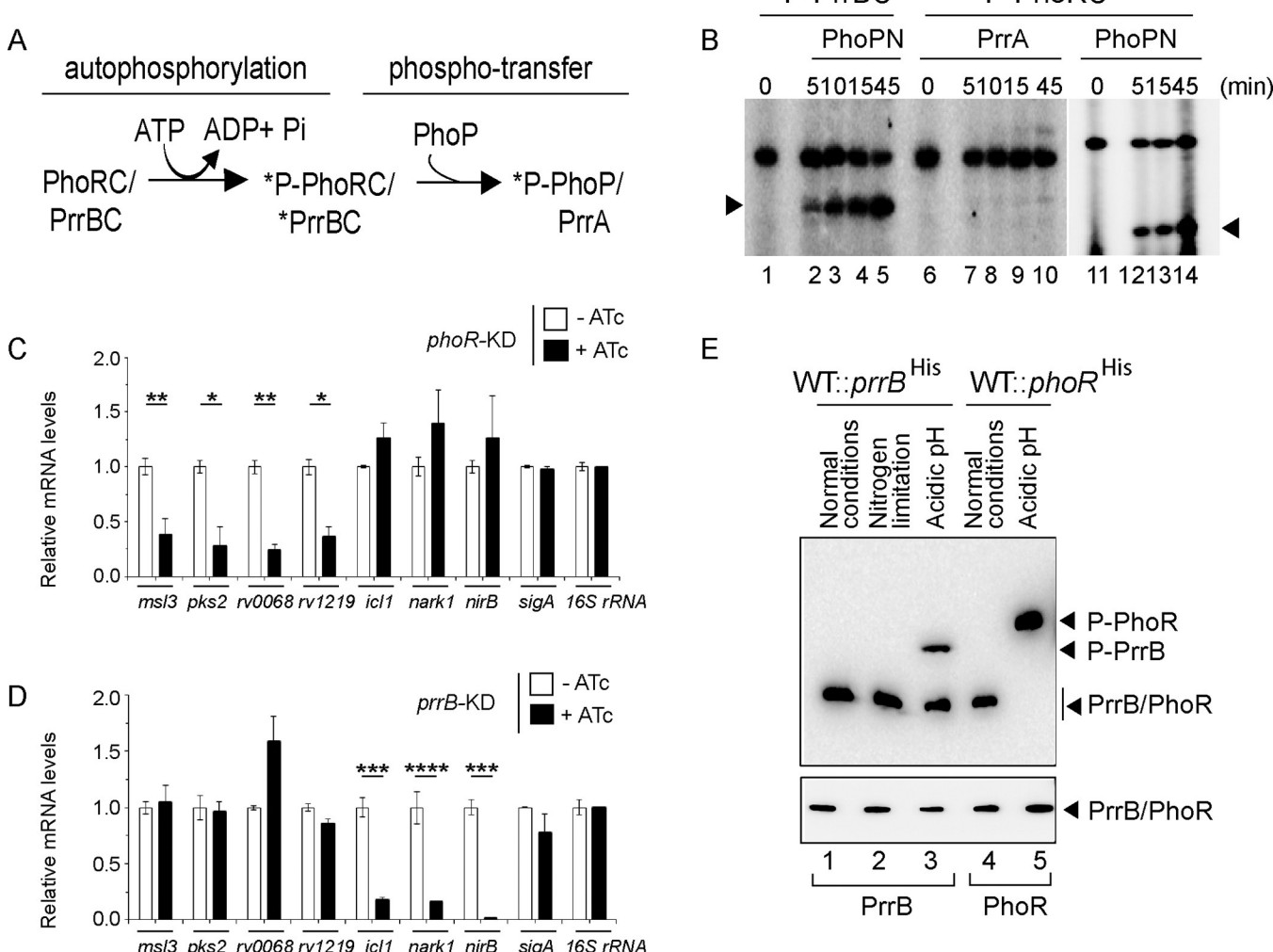

**Fig 6. A non-canonical pathway of phosphorylation of PhoP.** (A) Schematic diagram of auto-phosphorylation of SKs and subsequent phospho-transfer to RRs. (B) Phosphotransfer assays from radio-labelled PrrBC to PhoPN (lanes 1–5), PhoRC to PrrA (lanes 6–10), and PhoRC to PhoPN (lanes 11–15) are detailed in the Methods. Following SDS-PAGE analyses the reaction products were digitized by a phosphorimager (GE Healthcare); lane 1 and lane 6 resolve radio-labelled PrrBC and PhoRC, respectively. (C-D) mRNA levels of a few representative pH-inducible genes were next determined in the (C) *phoR* and (D) *prrB* knock-down (*phoR*-KD and *prrB*-KD, respectively) constructs, respectively, grown under low pH conditions of growth. Each value is an average of duplicate measurements originating from biological duplicates (*P<0.05; **P<0.01; ***P<0.001; ****P<0.0001). RT-qPCR measurements were performed as described in the Methods. (E) Phos-tag Western blot analysis of cell lysates prepared from WT-H37Rv harbouring His-tagged *prrB*. The bacterial strain was grown under normal conditions (pH 7.0) or indicated stress conditions, as described in the Methods, and the proteins of interest were detected by anti-His antibody. Data are representative of two independent experiments. As a loading control, His-PrrB or His-PhoR was detected from comparable amounts of cell lysates by Western blotting using anti-His antibody.

grown under nitrogen limiting conditions (lane 2). However, PrrB was partially (~50%) phosphorylated under acidic pH relative to normal conditions of growth (compare lanes 4 and 5). This observation is remarkably consistent with PrrB -dependent regulation of acidic pH-inducible gene expression (Fig 6D). In keeping with results shown in Fig 1, as a control, PhoR undergoes robust phosphorylation under acidic pH conditions of growth (compare lanes 4 and 5, Fig 6E). It should be noted that a significantly different gel composition of 8% SDS-PAGE (used in phosphotransfer assays; Fig 6B) versus 12.5% acrylamide including phos-tag (used for *in vivo* phosphorylation assays; Fig 6E) coupled with use of C-terminal fragment of the sensor kinases in the former experiment versus full length sensor proteins used in the

later, account for variable relative migration of phosphorylated sensor proteins. Together, these results uncover that acidic pH is also sensed in part, by *M. tuberculosis* PrrB, which in turn phosphorylates PhoP.

## PhoR is essential for mycobacterial survival in macrophages

Having shown that PhoR is essential for phosphorylation of PhoP as well as maintenance of P~PhoP homeostasis in a pH-dependent mechanism, we sought to investigate whether PhoR contributes to mycobacterial virulence. Using a mice infection experiment, a previous study by Wang and co-workers had shown that *M. tuberculosis* lacking a functional *phoR* displayed a lower lung burden of the bacilli relative to WT-H37Rv [33]. We used WT-H37Rv and *ΔphoR*-H37Rv to infect murine macrophages (Fig 7A). Although WT-H37Rv could effectively inhibit phagosome-lysosome fusion, *ΔphoR*- H37Rv readily matured into phagolysosomes, strongly indicating increased trafficking of the mycobacterial strain to lysosomes. To examine whether *ΔphoR*-H37Rv is growth defective or growth attenuated in macrophages, we compared survival of WT-H37Rv and *ΔphoR*-H37Rv within macrophages 3- and 48 -hrs post infection (Fig 7B). Notably, *ΔphoR*-H37Rv shows a comparable growth 3-hr post infection as that of

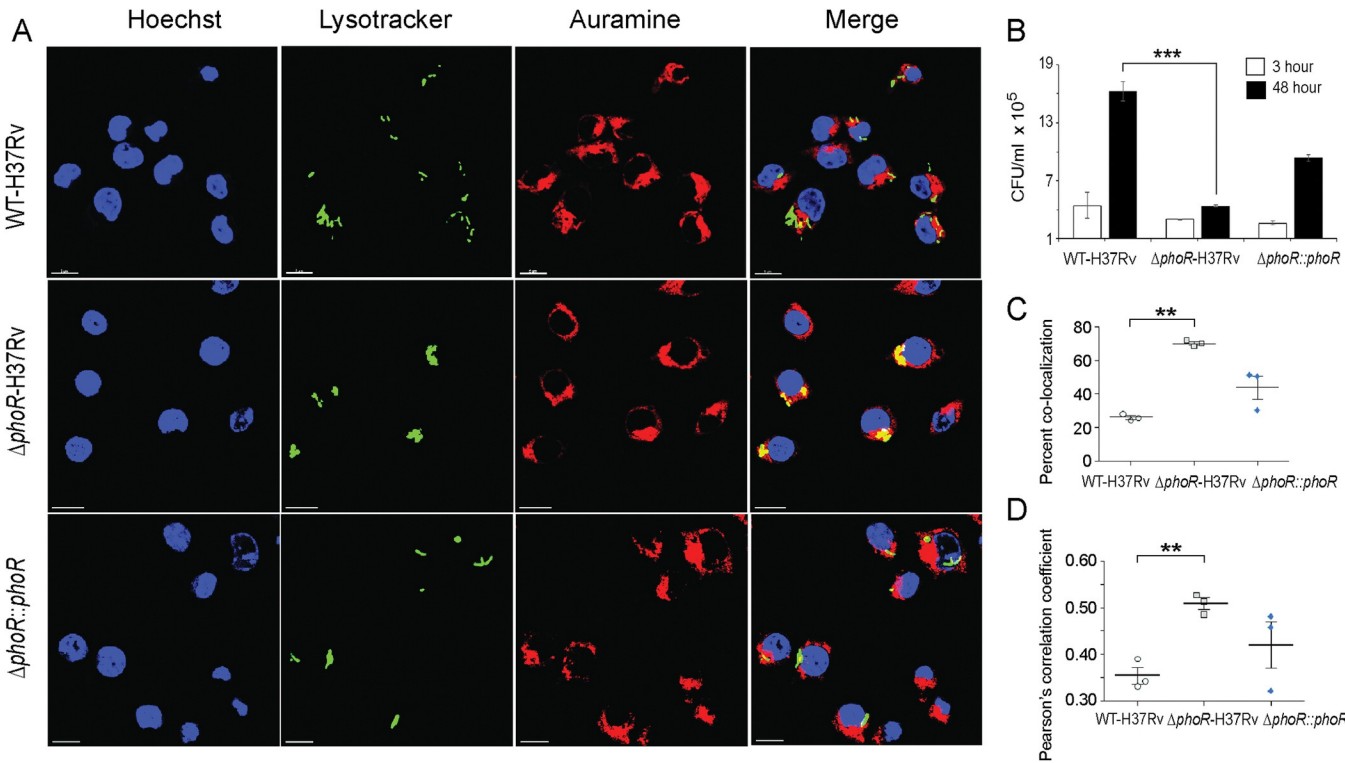

**Fig 7. PhoR contributes to mycobacterial survival in cellular models.** (A) WT-H37Rv and *ΔphoR*-H37Rv were used to infect murine macrophages. Mycobacteria and host cells were stained with phenolic auramine solution, and LysoTracker respectively. Host cell nuclei were made visible by Hoechst dye. Three fluorescence signals (Mycobacterial strains: green; lysosomes: red and host nuclei: blue) and their merging are displayed by confocal images (scale bar: 10 μm). (B) To examine contribution of PhoR to mycobacterial survival in cellular models, murine macrophages were infected with indicated mycobacterial strains, and 3- hour and 48-hour post infection intracellular bacterial CFU were enumerated. The results show average values from biological triplicates (***P<0.001). (C) Co-localization of auramine labelled mycobacterial strains with Lysotracker was investigated by visually scoring yellow and green punctas from at least 50 infected cells originating from 10 different fields of each of the three independent biological replicates. To determine percent co-localization, the number of yellow punctas were divided by the total number of punctas (yellow plus green) as described previously [72,73]. The results display average values from 50 infected cells (n = 50) of each independent experiment with standard deviations from three biological replicates (***P≤ 0.001). (D) The data present Pearson's correlation coefficient of images displaying internalized auramine-labelled mycobacteria and Lysotracker red marker in macrophages, and were evaluated using image-processing software NIS elements (Nikon). Average values with standard deviations were obtained from three independent experiments (*P<0.05; ***P<0.001).

WT-H37Rv. However, 48 hr post infection the mutant shows a significantly reduced growth (~3.7- fold) relative to WT-H37Rv. More importantly, growth-attenuation is significantly rescued in the complemented mutant strain, suggesting that *phoR* locus is essential for mycobacterial growth in macrophages. In keeping with these results, bacterial co-localization data (Fig 7C) and Pearson's plot (Fig 7D) strongly suggest that the *phoR* locus plays a major role to inhibit phagosome maturation.

These results are consistent with previously published animal infection data showing that a mutant *M. tuberculosis* strain carrying mutations around the primary phosphorylation site of PhoR displays a significantly lower lung burden of the bacilli relative to WT-H37Rv [33].

## Discussion

The first suggestion that *M. tuberculosis phoPR* system might be involved in pH sensing was made by Smith and co-workers, when they had reported PhoP-dependent regulation of numerous pH-sensitive genes in the transcriptome analyses [13]. Later, Russell and co-workers have shown a significant overlap between phagosomal acidic pH regulon and *phoPR* regulon [43]. These results led to the identification of a unique *M. tuberculosis* -specific acid and phagosome regulated *aprABC* locus [42]. Importantly, *aprABC* shows a *phoPR* -dependent activation during mycobacterial growth under acidic pH and contributes to remodel the phagosomal environment for effective intracellular adaptation of the bacilli [42]. The other suggested possibility was that *phoPR* responds to a non-conventional intra-bacterial signal associated with altered physiology at acidic pH, such as change in carbon metabolism and/or redox homeostasis [24,60]. Although we now know the *phoPR* regulon is induced under acidic pH *in vitro* and in macrophages [42,44], what remained unknown is whether PhoPR directly responds to low pH conditions.

While our present knowledge on the regulatory stimuli for a few SKs is often derived from transcriptomic data, for majority of SKs the environmental signals that directly regulate their activity remain unknown. This lack of knowledge accounts for our limited understanding on the contribution of these systems to different host niches in the context of mycobacterial pathogenesis. In this study, we observed a significantly elevated level of phosphorylation of PhoR under acidic pH, and high salt conditions of growth relative to normal conditions (Fig 1A). Importantly, under identical stress conditions we observed a significantly elevated phosphorylation of PhoP (Fig 1B), suggesting a stress-specific likely phospho-transfer from PhoR to PhoP. Further, we demonstrated that P~PhoP is almost undetectable in *ΔphoR*-H37Rv even under acidic pH (Fig 2), allowing us to conclude that acidic pH/high salt condition is directly sensed by PhoR, and low pH promotes PhoR-dependent PhoP phosphorylation. Consistent with these results, transcriptome analyses of *ΔphoR*-H37Rv and WT-H37Rv demonstrate a strikingly differential regulation of low pH-inducible PhoP regulon with a significant impact of PhoR on *in vivo* recruitment of PhoP within acidic-pH inducible promoters (Fig 3). Collectively, these results suggest that lack of acidic pH- inducible expression of PhoP regulon in *M. tuberculosis* is attributable to the absence of PhoR.

To investigate regulation of P~PhoP in mycobacteria, we next discovered that PhoR functions as a robust phosphatase of P-PhoP, suggesting that the sensor in addition to its kinase function undertakes an additional regulatory function to inhibit 'triggering ON' of the PhoP regulon unless it is necessary. This is not an unexpected result as *M. tuberculosis* PhoR is known to exhibit autophosphatase activity [61] and phosphatase activity of SK controlling functionality of the cognate RR has been reported for *M. smegmatis senX3-regX3* signalling system [62]. However, the finding that PhoR is a dual function sensor protein with both kinase and phosphatase functions (Figs 1 and 4) provides an explanation to how PhoP regulon

remains repressed under non-inducing conditions. We consider two major physiological significances of the newly-identified function of PhoR. First, it remains an effective mechanism to prevent PhoP regulon expression which perhaps is only needed during mycobacterial adaptation to low pH. Secondly, restoring PhoP to its unphosphorylated status by PhoR may be essential to reverse pH regulon under normal conditions that support increased metabolic activity. It is noteworthy that PhoP plays a major role in integrating low pH conditions and redox homeostasis by controlling mycobacterial metabolic plasticity [24,60]. Thus, PhoR controls the net phosphorylation status of PhoP and determines the final output on mycobacterial adaptive programme to low pH conditions, most likely through its contrasting kinase and phosphatase activities, facilitates an integrated view of our results. The balancing act of two activities, which controls the expression of acidic pH-inducible PhoP regulon is schematically shown as a model in Fig 4J. The result identifying E260, adjacent to the phosphorylation site H259, as a critically important residue for PhoR phosphatase function (Fig 4H), provides us with a fundamental biological insight into how a limited number of residues of the DHp domain contribute to two contrasting functions of the sensor protein and determine environment-specific net activation status of the cognate RR. Importantly, PrrB comprises of an identical consensus phosphatase motif (amino acid residues 240–246) as that of PhoR. However, BLAST search fails to pick up PrrB as one of the closest homologues of PhoR possibly because of lack of sequence homology and/or identity around the active site (Fig 4F). Perhaps, these residues around the active site define specificity of phosphatase activity because of which despite being able to phosphorylate PhoP (Fig 6B), PrrBC was unable to dephosphorylate P~PhoP (Fig 4D). Collectively, these results suggest an additional tier of regulation of homeostatic mechanism of P~PhoP.

While conserved structural elements of TCS signalling systems are significantly similar, growing evidence suggests interactions between TCSs. To explore a possible non-canonical mechanism of activation of PhoP, we examined cross-talk between PhoP and other mycobacterial SKs. These cross-talks, which were defined as communication between two functional pathways, are now referred to as cross-regulation requiring an activating signal [63]. While only a very few examples of natural cross regulations are known, reciprocal signalling between two TCSs (HssRS-HitRS) was demonstrated as an elegant example of bacterial signal cross-regulation integrating response to heme and cell envelope stress for an effective adaptation of *Bacillus anthracis* to the vertebrate host [64]. Also, numerous studies were undertaken to investigate cross-talk between mycobacterial TCS proteins [51–53]. Although these studies suggest "multi-to-one" signalling relationships, the physiological consequences of these interactions to environmental adaptations, or how they facilitate important cellular functions remain unknown. In recent years a number of regulators have been shown to interact with PhoP impacting mycobacterial physiology [21,23,25,44]. However, till date phosphorylation of PhoP with a non-cognate kinase have not been documented. As we begin to understand the contribution of TCS proteins to mycobacterial physiology beyond the classical phospho-transfer pathway, it becomes evident that these systems participate in extensive signalling networks. These pathways may have important consequences on pathogenesis particularly in view of the fact that many of the environmental conditions encountered by *M. tuberculosis* are part of the same niche, suggesting that cross-regulation between concomitantly activated TCSs may be one of the common mechanisms pathogenic mycobacteria employ to facilitate environmental adaptation. To investigate whether PhoP is activated by any other SK(s), using M-PFC experiments (Fig 5) coupled with phosphorylation assays (Fig 6), we demonstrate specific cross-talk of PhoP with PrrB. Although these results appear to be in keeping with the role of *phoP* locus in mycobacterial nitrogen metabolism [21], we were unable to detect *in vivo* phosphorylation of PrrB (Fig 6E) under nitrogen limiting conditions. However, we unexpectedly observed PrrB

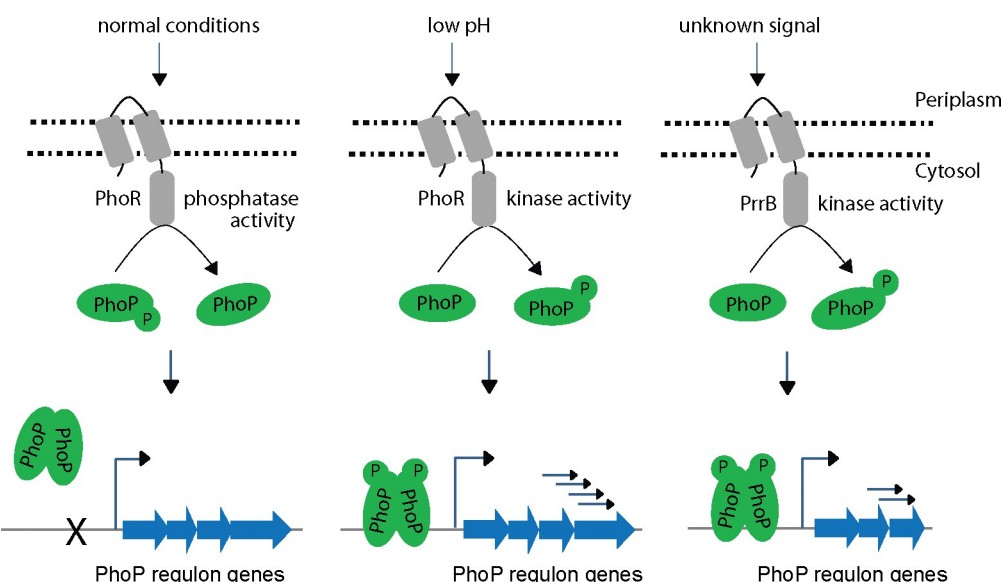

**Fig 8. Schematic summary of canonical and non-canonical mechanisms of stress-specific phosphorylation of PhoP.** Indicated sensor kinases upon sensing stress conditions, phosphorylate PhoP, and subsequently P~PhoP bind to its target promoters to activate the regulon under its control.

undergoing partial phosphorylation *in vivo* under acidic pH conditions of growth relative to normal pH (compare lanes 1 and 3, Fig 6E). Together, this counterintuitive finding strikingly accounts for PhoR-independent, but PhoP-mediated regulation of a few pH-inducible genes (Figs 3A, 6C, 6D and S5B). Note that the environmental signal(s) that triggers specific and non-canonical mode of PhoP activation via PrrB, requiring an appropriate transcriptional response, remains unknown. Although we show that PhoP is phosphorylated by PrrB (Fig 6B) and PrrB is partially phosphorylated under acidic pH (Fig 6E), we were unable to conclusively demonstrate that PrrB phosphorylates PhoP in response to acidic conditions. In fact, we do not think PrrB functions as a sensor of acidic pH, rather we speculate that PrrB is partly phosphorylated in response to a signal that mimics altered physiology under acidic pH, for the following reasons. First, recombinant PrrBC in a robust reaction can phosphorylate PhoPN *in vitro* (Fig 6B). Second, in the mycobacterial cell lysates while 100% of the PhoR protein was present in the phosphorylated form, only ~35% (based on the limits of detection in this assay) PrrB becomes phosphorylated *in vivo* under identical acidic conditions of growth (Fig 6E). Third, and most importantly, a set of acidic pH -inducible PhoP regulon genes are significantly down-regulated in a *prrB*-KD mutant, but not in a *phoR*-KD strain (Fig 6C and 6D). Together, these results, for the first time, connect two SKs, PhoR and PrrB, to a single RR (PhoP) in a stress-specific manner and provide new biological insights into mechanism of activation of the versatile regulator (see Fig 8).

Further, these results uncover that a SK from one TCS communicates with a non-cognate RR of another TCS to expand the signalling repertoire of mycobacteria. In this "multi-to-one" signalling pathway, promiscuous SKs phosphorylate more than one RRs, impact input recognition capabilities of RR and subsequently expand and/or fine tune downstream response by the phosphorylated RR. We propose that the multi-stimulus SK-dependent regulation of a RR likely reflects the need of the pathogen to fine-tune its complex physiology within the colonized niche. It is possible that rather than an isolated exception, RRs getting activated in response to multiple signals is not so uncommon. In fact, fewer TCSs of *M. tuberculosis* relative

to other bacterial species of comparable genome size is suggestive to be part of a more efficient regulatory system. With more insights appreciating host adaptation of the tubercle bacilli which rely on bacterial TCSs—is an area of growing interest that holds strong potential to develop adjunct therapy. It is noteworthy that ethoxzolamide, which inhibits PhoPR virulence associated regulon in *M. tuberculosis* [65], has been shown to reduce bacterial burden in both infected macrophages and mice. Although the mechanism is yet to be understood, ethoxzolamide is thought to inhibit PhoR sensing by targeting cell surface carbonic anhydrases.

In conclusion, our results for the first time demonstrate that acidic pH activates PhoP by promoting its phosphorylation in a PhoR-dependent manner, and therefore, expression of PhoP regulon requires the presence of PhoR. We also identify a non-canonical mechanism of activation of PhoP, connecting two SKs with signal dependent activation of the virulence regulator. Our results further uncover phosphatase activity of PhoR, and identify the motif and residues responsible for kinase/phosphatase (dual) functioning of the SK. Collectively, we conclude that both kinase and phosphatase functions of PhoR remain critically important for downstream functioning of the activated regulator, and propose a model suggesting how dual functioning of PhoR effectively regulates final output of the PhoP regulon in an environment-dependent manner. In keeping with these results, PhoR remains essential for mycobacterial virulence, and together, these results have striking implications on the mechanisms of virulence regulation.

## Materials and methods

### Bacterial culture, and growth experiments

*E. coli* DH5α, and *E. coli* BL21 (DE3) were utilized to clone and express mycobacterial proteins, respectively. Construction of PhoR-deleted *M. tuberculosis* (*ΔphoR*-H37Rv) using 'mycobacterial recombineering' and complementation of the mutant have been described in this study. PhoPR-deleted *M. tuberculosis* (*ΔphoP*-H37Rv) and the complemented mutant have been described previously [13]. *M. tuberculosis* strains were grown aerobically in 7H9 liquid broth or on 7H10-agar medium as described [30]. Wild-type (WT) or mutant *M. tuberculosis* strains were transformed and selected in presence of antibiotics as described [66]. For stress conditions due to low pH and salt, cells were grown to $OD_{600}$ of 0.4, washed with 7H9 buffered with 2N HCl for pH 7.0 or pH 4.5, or 7H9 containing 250 mM sodium chloride, respectively, re-suspended in media of indicated pH and salt, and grown for additional 2 hours at 37˚C. For redox stress, mycobacterial cells were inoculated into 7H9-ADS (albumin-dextrose-sodium chloride) containing ~5 mM diamide (Sigma) at $OD_{600}$ of 0.05, and grown for additional 48 hours at 37˚C as described [67]. For mycobacterial growth under nitrogen limiting conditions, cells were grown in synthetic 7H9 medium using 0.04 g/L ferric citrate and 0.5 g/L sodium sulphate instead of comparable concentrations of ferric ammonium sulphate, ammonium sulphate and 0.5 g/L glutamic acid, as nitrogen sources.

### Cloning

His-tagged PhoP was cloned between BamHI and PstI sites of both p19Kpro [37] and pSTKi [68] using the same primer pair FPmphoP/RPmphoP (S1 Table). Likewise, full-length PhoR, and PrrB were cloned in pSTKi and p19Kpro between NdeI/PstI and BamHI/HindIII sites, using primer pairs FPphoRcom/RPphoRcom and FPprrB/RPprrB, respectively. *M. tuberculosis* PhoP, PhoR, and PrrB proteins carrying an N-terminal $His_6$-tag were expressed in mycobacteria as described previously [23]. Cloning of cytoplasmic domain of PhoR (PhoRC comprising residues 193–485 of PhoR), full-length DosR, and PhoP and expression in *E. coli* have been described earlier [21,34]. To express $His_6$-tagged cytoplasmic domain of PrrB

(PrrBC, comprising residues 200–446 of PrrB), the amplicon was inserted in pET28b between NdeI and HindIII sites, and PrrBC was expressed as described for PhoRC [34]. Likewise, NdeI and HindIII sites of pET28b (Novagen) were used to construct plasmid expressing PrrA. Specific point mutations within the indicated ORFs were introduced by overlap extension PCR, and constructs were checked by DNA sequencing. The sequences of the oligonucleotides used for cloning and the plasmids used for expression are listed in S1 and S2 Tables, respectively.

### Expression and purification of proteins

Plasmids expressing recombinant WT or mutant PhoP, DosR, PrrA, cytoplasmic domains of PhoR and PrrB proteins, were expressed as fusion proteins containing N-terminal His$_6$- tag and purified by Ni-NTA chromatography as described [34,69]. In each case, protein purity was examined by SDS-PAGE, and concentration was assessed by Bradford reagent (with BSA as a standard). The storage buffer for proteins comprised of 50 mM Tris-HCl, pH 7.5, 300 mM NaCl, and 10% glycerol.

### *In vitro* phosphorylation of proteins

For autophosphorylation, WT or mutant PhoRC, and PrrBC proteins (~ 3 μM) were included in a phosphorylation mix (50 mM HEPES, pH 7.5, 50 mM KCl, 10 mM MnCl$_2$) which comprised of 25 μM of (γ-$^{32}$P) ATP (BRIT, India). The reactions were allowed to continue for 1 hr at 25˚C. To initiate phosphotransfer, ~ 3 μM PhoP or its truncated variants were included in the phosphorylation, the reactions continued at 18˚C for indicated times, and terminated by 10 mM EDTA. The reactions products were analysed by SDS/polyacrylamide gels, gels were dried, and signals digitized by a phosphorimager.

### *In vivo* protein phosphorylation experiments

Cell lysates were resolved in polyacrylamide gels containing acrylamide–Phos-tag ligand (Wako Laboratory, Japan) as per manufacturer's recommendation to detect PhoP/PhoR proteins and their phosphorylated forms. These gels were copolymerized in presence of 50 μM Phos-tag acrylamide and 100 mM MnCl$_2$. *M. tuberculosis* cell-lysates were prepared as described previously [23] and total protein content determined by Bradford reagent. Standard running buffer [0.4% (w/v) SDS, 25 mM tris, 192 mM glycine] was used for electrophoresis of samples on Phos-tag gels for 5 hours under 20 mA. Next, the resolved samples were transferred to PVDF membrane and detected by Western blot analyses. Anti-His antibody was from GE Healthcare; anti-PhoP antibody was elicited in rabbit (AlphaOmegaSciences); goat anti-rabbit and goat anti-mouse secondary antibodies were from Abexome Biosciences, and the chemiluminescence reagent to develop the blots was from Millipore.

### Construction of *ΔphoR*- H37Rv

PhoR-deleted *M. tuberculosis* H37Rv was constructed by 'mycobacterial recombineering' [39]. To construct *phoR* allelic exchange substrate (AES), left homology region (LHR) encompassing -970 to +30 (relative to *phoR* ORF) and the right homology region (RHR) encompassing +1423 to +2423, were PCR amplified using the primer pairs FPphoRLHR/ RPphoRLHR and FPphoRRHR/ RPphoRRHR, respectively. The primers included PflM1(LHR) and DraIII (RHR) restriction sites, and digestion with these enzymes results in ends that are compatible for cloning with Hyg$^r$ cassette (derived from modified-pENTR/D-TOPO plasmid) and OriE + cosλ fragment (derived from p0004S plasmid). Also, ScaI site was inserted both in the LHR forward primer and RHR reverse primer. Next, amplicons of LHR and RHR were digested

with PflM1 and DraIII, respectively; and the resultant fragments were ligated with Hyg$^r$ cassette (1.3 kb) and OriE+ cosλ fragment (1.6 kb) to generate *phoR* AES. Next, it was digested with ScaI to generate linear *phoR* AES (~3.3 kb) suitable for 'recombineering'. Fig 2A shows the schematic map of *phoR* AES plasmid.

For homologous recombination, WT-H37Rv strain was electroporated with pNit-ET (a kind gift of Prof. Eric Rubin) expressing gp60 and gp61 to generate recombineering proficient H37Rv::pNit-ET strain. This strain was grown to $OD_{600} \approx 0.4$ and 0.5 μM isovaleronitrile (Sigma) induced expression of the recombinases. Next, 200 ng of *phoR* linear AES was used to electroporate cells and transformed cells were grown on to 7H10/OADC agar plates containing hygromycin. Genomic DNAs were isolated from single colonies and screened for *phoR* replacement. To construct the complemented mutant, *phoR* encoding ORF was PCR amplified using the primer pair FPphoRcomp/ RPphoRcomp, digested with NdeI/PstI and ligated to double digested pST-Ki [68]. Next, pST-*phoR* clone was confirmed by sequencing, electrotransformed in *ΔphoR*-H37Rv competent cells, and the transformed colonies were selected on 7H10/OADC/Kan/Hyg. The mutant and the complemented strains were verified by gene-specific PCR amplification using corresponding genomic DNAs as PCR templates. We also performed Southern blot hybridization to confirm the mutant, and RT-qPCR experiments compared *phoR* mRNA levels of WT-H37Rv, mutant, and the complemented strain.

## Southern blot hybridization

This approach was used to confirm *ΔphoR*-H37Rv mutant. Approximately 1 μg of genomic DNA of WT and *ΔphoR*-H37Rv were restricted using BamHI and separated by agarose gel electrophoresis. Next, the DNA was transferred onto the Immobilon membrane (Millipore) by vacuum-based blotting, and hybridized with radio-labelled gene-specific probes. The *hrcA* and *phoR*-specific probes (S1 Table) were generated by PCR using alpha $^{32}$P-dCTP (BRIT, India) and oligonucleotide primers, which were used to clone the respective ORFs. Pre-hybridization, hybridization and washing steps were in accordance with the standard protocol. The image was developed and digitized with a phosphorimager (GE Health care).

## Construction of *phoR* and *prrB* knock-down mutants of *M. tuberculosis* H37Rv

In this study, we utilized a previously-described CRISPRi -based strategy [57] to construct knock-down mutants of *phoR* (*phoR*-KD) and *prrB* (*prrB*-KD). This approach inhibits expression of genes via inducible expression of dCas9 protein using target gene-specific guide RNAs (sgRNA). First, WT-H37Rv was transformed with *S. pyogenes* dCas9 expressing integrative plasmid pRH2502 to generate *WT-H37Rv::dCas9*. Next, 20-nt long *phoR* and *prrB*-targeting spacer sgRNAs were cloned in pRH2521 using BbsI enzyme and the constructs sequenced. The oligonucleotides were designed such that the expressed sgRNA comprises of a 20 bp sequence that remains complementary to the non-template strand of the target gene. Finally, the corresponding clones were electroporated into *M. tuberculosis* harbouring pRH2502. To express dcas9 and repress sgRNA-targeted genes (*phoR* or *prrB*), the bacterial cultures were grown in Middlebrook 7H9 broth, supplemented with 0.2% glycerol, 10% OADC, 0.05% Tween, 50 μg/ml hygromycin and 20 μg/ml kanamycin at 37°C, anhydrotetracycline (ATc) was added every 48 hours to a final concentration of 600 ng/ml, and cultures were continued to grow for 4 days. Next, RNA isolation was carried out, and RT-qPCR experiments verified repression of target genes. For the induced strains (in the presence of ATc) expressing sgRNAs targeting +234 to +253 (relative to *phoR* translational start site) and +56 to +75 sequences (relative to *prrB* translational start site), we obtained approximately 92% and 95% reduction of

*phoR* and *prrB* RNA abundance, respectively, compared to the respective un-induced strains. The oligonucleotides used to generate gene-specific sgRNA constructs and the plasmids utilized in knock-down experiments are shown in S1 and S2 Tables, respectively.

## RNA isolation

Isolation and purification of RNA from *M. tuberculosis* strains grown under normal and varying conditions, were carried out as described previously [31]. To ensure removal of genomic DNA, each RNA preparation was incubated with RNase-free DNaseI at room temperature for 20 minutes. RNA concentrations were measured by recording absorbance at 260 nm, and sample integrity was verified using formaldehyde-agarose gel electrophoresis by assessing intactness of 23S and 16S rRNA.

## RNA sequencing and data analysis

RNA-sequencing and data analyses have been performed by AgriGenome and MedGenome Lab Ltd. (India). Agilent 2200 system examined RNA integrity, and 'si-tools Pan-prokaryote ribopool probes removed rRNA. TruSeq standard RNA library prep kit was used to prepare the libraries, which were sequenced by Illumina HiSeq x10 platform to generate 150-bp paired-end reads. Details of data analyses was as described [67]. Fold changes $\geq$1.5 or $\leq$-1.5 of treated (compared to control) samples were used to generate heat-maps using GENE-E software. Our results showing a comprehensive list of genes have been submitted in the NCBI's database (GEO accession number GSE180161).

## Quantitative Real-Time PCR

Total RNA extracted from *M. tuberculosis* cultures grown under specific conditions were used for cDNA synthesis and PCR reactions using Superscript III platinum-SYBR green one-step RT-qPCR kit (Invitrogen). Details of PCR conditions are described previously [67], and *M. tuberculosis* RpoB or 16S rDNA was used as endogenous controls. Two independent RNA preparations were always used to evaluate fold difference in gene expression using $\Delta\Delta C_T$ method [70]. S5 Table lists sequences of oligonucleotide primers, which were utilized to determine specific mRNA levels relative to control set displaying no differential expression. Enrichment attributable to PhoP binding to its target promoters were assessed by using 1 μl of IP or mock IP (no antibody control) DNA with SYBR green mix (Invitrogen) and promoter-specific primers.

## ChIP-qPCR

Details of ChIP-qPCR experiments using anti-PhoP antibody has been described elsewhere [45]. *In vivo* recruitment of the regulator was assessed using IP DNA in a reaction buffer containing SYBR green mix (Invitrogen), appropriate PAGE-purified primers and one unit of Platinum Taq DNA polymerase (Invitrogen). Amplifications were done for 40 cycles using Applied Biosystems real-time PCR detection system, and signal from an IP without antibody was compared to measure efficiency of recruitment. PCR-enrichment specificity from identical IP samples was also verified using 16S rDNA or RpoB- specific primers. A single product was amplified in all cases, and duplicate measurements were made in each case from two independent bacterial cultures.

## Mycobaterial protein fragment complementation (M-PFC) assays

*M. tuberculosis phoP* was expressed from pUAB400 (Kan^R), and transformed cells (*M. smegmatis* mc^2155 containing pUAB400-*phoP*) were grown in liquid medium to obtain competent

cells. Likewise, sensor kinases were expressed from the episomal plasmid pUAB300 (Hyg$^R$) after cloning their ORFs between BamHI/HindIII sites. Sequences of the oligonucleotide primers and relevant plasmids are listed in S6 and S7 Tables, respectively. Each construct was checked by DNA sequencing, and M. smegmatis cells carrying both plasmids were selected on 7H10/Kan/Hyg plates in presence or absence of 15 μg/ml Trimethoprim (TRIM) as described earlier [54]. *phoP/phoR* co-expressing constructs were used as a positive control [71].

## Macrophage infections

RAW264.7 macrophages were infected with titrated cultures of H37Rv, *ΔphoR*-H37Rv and complemented mutant strain at a multiplicity of infection (MOI) of 1:5 or 1:10 for as described previously [23]. While phenolic auramine solution was used to stain *M. tuberculosis* H37Rv strains, the cells were stained with 150 nM Lyso-Tracker Red DND 99 (Invitrogen). Next, cells were fixed, analysed using confocal microscope (Nikon, A1R), and processing of digital images were carried out with IMARIS imaging software (version 9.20). Details of the experimental methods and the laser/detector settings were optimized using macrophage cells infected with WT-H37Rv as described [23]. A standard set of intensity threshold was made applicable for all images, and percent bacterial co-localization was determined by analyses of at least 50 infected cells originating from 10 different fields of each of the three independent biological repeats.

## Supporting information

**S1 Fig. Confirmation of PhoR-deleted *M. tuberculosis* H37Rv.** (A) Gene-specific PCR reactions utilized genomic DNA of WT-H37Rv and *ΔphoR*-H37Rv and a pair of *phoR*-specific primers (FPphoRInt/RPphoRInt), and products were resolved on agarose gel. Although WT-H37Rv yields a ~0.6 kb- *phoR*-specific product (lane 1), genomic DNA of *ΔphoR*-H37Rv fails to yield a *phoR*- specific amplicon (lane 2). However, the complemented mutant (lane 3) harbouring a copy of *phoR* shows the presence of the specific amplicon; lane 4, DNA molecular weight marker. As a positive control, *phoP*-specific amplicon was present in the three strains (lanes 5–7). (B) Southern blot analyses using genomic DNA of WT-H37Rv showed two specific bands when probed with end-labelled *hrcA* and *phoR*-specific probes (approximately 9.6- and 2-kb, respectively). However, *ΔphoR*-H37Rv genomic DNA, using identical probes could detect *hrcA*-specific band (~9.6 -kb), but not the *phoR*-specific product.
(TIF)

**S2 Fig. Global expression of acidic pH-inducible genes in *M. tuberculosis*.** (A-B) RNA-sequencing derived heat-map showing 167 significantly activated acidic pH-inducible genes of WT-H37Rv (>1.5 -fold, p< 0.05, q<0.05) grown under low pH (pH 4.5) relative to normal conditions (pH 7.0) of growth. (C) Expression profile of acidic pH-dependent downregulated mycobacterial genes. RNA-sequencing derived heat-map showing ~ 22 low pH-dependent down-regulated genes in *ΔphoR*-H37Rv (>1.5 fold; p< 0.05) grown under low pH (pH 4.5) conditions compared to normal conditions (pH 7.0) of growth. The data represent average of biological replicates, and list acidic pH-dependent significantly downregulated genes of the mutant. (D-E) The Volcano plots display differential expression of genes (> 1.5- fold, p< 0.05, q<0.05) grown under low pH (pH 4.5) relative to normal conditions (pH 7.0) of growth in case of (D) WT-H37Rv and (E) *ΔphoR*-H37Rv, respectively. In these plots, each dot represents expression level of a gene; while red and green dots indicate statistical significance, black dots represent lack of statistical significance. Of the differentially expressed genes, 31 genes were significantly up-regulated (Fig 3A), and 22 genes were significantly down-regulated (S2C Fig) in *ΔphoR* -H37Rv as a function of acidic pH. Note that under identical conditions 167 genes

showed upregulation whereas 89 genes were downregulated in WT-H37Rv.
(TIF)

**S3 Fig. Expression of *phoR*-independent acid-inducible mycobacterial genes.** (A-C) PhoR-independent expression of acidic pH-inducible genes were examined by comparing expression of representative genes in (A) *ΔphoR*-H37Rv, (B) WT-H37Rv and (C) complemented *ΔphoR*-H37Rv strain grown under normal and acidic pH conditions. The data show plots of average values from two biological duplicates, each performed with one technical repeat (*P≤0.05; **P≤0.01). (D) To examine *in vivo* recruitment of PhoP within its target promoters, FLAG-tagged PhoP was expressed in *ΔphoR*-H37Rv, and ChIP was carried out using anti-FLAG anti-body followed by qPCR using IP samples (see Methods section for further details). To assess fold enrichment, each data point was compared with the corresponding IP sample without adding antibody. The experiments were carried out as biological duplicates, each with one technical repeat (*P≤0.05).
(TIF)

**S4 Fig. Mutations in the DHp domain of *M. tuberculosis* PhoR.** (A) The structural model of PhoRC DHp domain utilized structural coordinates of the SK [PDB ID: 5UKY] and was generated by Pymol (http://www.pymole.org/). The two most conserved residues are indicated on the figure. (B) Four recombinant PhoRC mutants, each carrying a conservative substitution of a single amino acid residue of PhoR DHp domain, were cloned, expressed and purified as described in the Materials and Methods. Purified proteins (≈2 μg/lane) were analyzed by SDS/polyacrylamide gel electrophoresis and visualized by Coommassie blue staining. The sizes of the molecular mass markers (in kDa) are indicated to the left of the figure. See 'Results' section for a description of the PhoRC mutants. (C) The indicated recombinant mutants were purified and auto-phosphorylated as described in the methods. The products were resolved by SDS-PAGE and digitized by a phosphorimager.
(TIF)

**S5 Fig. Confirming sensor-kinase knock-down mutants.** (A) Expression levels of indicated genes in *phoR* and *prrB* knock-down constructs grown under acidic pH (pH 4.5) conditions of growth. To compare respective mRNA levels, gene-specific expression was determined by RT-qPCR (see Methods). The results display average values derived from biological duplicates, each performed with two technical repeats (*P≤0.05; **P≤0.01; ***P≤0.001). (B) To compare relative expression of a few representative genes, RNA-seq data from WT-H37Rv and *ΔphoR*-H37Rv, grown under acidic pH conditions of growth, were analysed and shown as a heat-map.
(TIF)

**S1 Table. Oligonucleotide primers used for cloning and amplifications reported in this study.**
(DOCX)

**S2 Table. Plasmids used for cloning and expression reported in this study.**
(DOCX)

**S3 Table. RNA-seq derived gene expression data of WT-H37Rv and *ΔphoR*-H37Rv grown under normal conditions (pH 7.0) and acidic pH (pH 4.5).**
(XLSX)

**S4 Table. Activation and repression of gene expression in indicated mycobacterial strains grown under acidic pH (pH 4.5) versus normal conditions of growth (pH 7.0).**
(XLSX)

**S5 Table. Sequences of oligonucleotides utilized in RT-PCR and ChIP experiments reported in this study.**
(DOCX)

**S6 Table. Sequences of oligonucleotides used in M-PFC experiments reported in this study.**
(DOCX)

**S7 Table. Plasmids used in M-PFC experiments reported in this study.**
(DOCX)

## Acknowledgments

We acknowledge G. Marcela Rodriguez and Issar Smith (PHRI, New Jersey Medical School—UMDNJ) for Δ*phoP*-H37Rv, and the complemented mutant strain. We thank Divya Arora, Vinay Nandicoori, and Ritesh Sevalkar for their help in constructing Δ*phoR*-H37Rv. We acknowledge Rajni Garg, and Anunay Sinha from the laboratory of Sanjeev Khosla for their help in constructing the knock-down mutants, and analysis of RNA sequencing data, respectively.

## Author Contributions

**Conceptualization:** Prabhat Ranjan Singh, Partha Paul, Dibyendu Sarkar.

**Data curation:** Prabhat Ranjan Singh, Harsh Goar, Partha Paul, Khushboo Mehta, Bhanwar Bamniya, Anil Kumar Vijjamarri, Roohi Bansal, Hina Khan, Subramanian Karthikeyan, Dibyendu Sarkar.

**Formal analysis:** Prabhat Ranjan Singh, Harsh Goar, Partha Paul, Khushboo Mehta, Bhanwar Bamniya, Dibyendu Sarkar.

**Funding acquisition:** Dibyendu Sarkar.

**Investigation:** Prabhat Ranjan Singh, Harsh Goar, Partha Paul, Khushboo Mehta, Bhanwar Bamniya, Anil Kumar Vijjamarri, Roohi Bansal, Hina Khan, Subramanian Karthikeyan, Dibyendu Sarkar.

**Methodology:** Prabhat Ranjan Singh, Harsh Goar, Partha Paul, Khushboo Mehta, Bhanwar Bamniya, Anil Kumar Vijjamarri, Roohi Bansal, Hina Khan.

**Project administration:** Dibyendu Sarkar.

**Resources:** Prabhat Ranjan Singh, Harsh Goar, Partha Paul, Khushboo Mehta, Bhanwar Bamniya, Anil Kumar Vijjamarri.

**Software:** Prabhat Ranjan Singh, Harsh Goar, Partha Paul, Khushboo Mehta, Subramanian Karthikeyan.

**Supervision:** Dibyendu Sarkar.

**Visualization:** Harsh Goar, Partha Paul, Khushboo Mehta, Bhanwar Bamniya.

**Writing – original draft:** Dibyendu Sarkar.

**Writing – review & editing:** Partha Paul, Khushboo Mehta, Bhanwar Bamniya, Dibyendu Sarkar.

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
