## [Decision Letter · Decision Letter 0]

10 Jul 2023

Dear Dr Sarkar,

Thank you very much for submitting your Research Article entitled 'Dual functioning by the PhoR sensor is a key determinant to Mycobacterium tuberculosis virulence' to PLOS Genetics.

The manuscript was fully evaluated at the editorial level and by independent peer reviewers. The reviewers appreciated the attention to an important problem, but raised some substantial concerns about the current manuscript. Based on the reviews, we will not be able to accept this version of the manuscript, but we would be willing to review a much-revised version. We cannot, of course, promise publication at that time.

If you decide to revise the manuscript for further consideration at PLOS Genetics, please aim to resubmit within the next 60 days, unless it will take extra time to address the concerns of the reviewers, in which case we would appreciate an expected resubmission date by email to plosgenetics@plos.org.

We are sorry that we cannot be more positive about your manuscript at this stage. Please do not hesitate to contact us if you have any concerns or questions.

Yours sincerely,

Danielle A. Garsin

Academic Editor

PLOS Genetics

Sean Crosson

Section Editor

PLOS Genetics

Reviewer's Responses to Questions

**Comments to the Authors:**

Reviewer #1: The manuscript “Dual functioning by the PhoR sensor is a key determinant to Mycobacterium tuberculosis virulence” by Singh et al, revisits a very important question in the mycobacterial TCS crosstalk that have from time to time shown to play a very important role in adaptation and survival of the bacteria to various stresses. In this work the authors have shown that the PrrB can acct as a non-canonical phosphate donor to PhoP. They have also explored the dual function (kinase and phosphatase) of the PhoR.

First, I would like to say that the paper is very well written, especially mentioning the lanes in text is highly appreciable as it makes it easy for the readers to understand. The labelling of the figures is also very good. My only suggestion would be to include the non-cognate PrrB axis later in the paper just before the macrophage and mice experiment. This will help the readers to keep up with the flow. Despite some interesting observations, given below are some of my comments and questions that must be addressed before publication.

Major comments:

1: It is already known that PhoP is instrumental in acid stress (Bansal et al 2017) and PhoR is the cognate partner of PhoP. From Fig 1 the takeaway message is that PhoR is getting phosphorylated at acidic condition and in absence of PhoR there is no *P-PhoP seen. Yes, previous studies have either worked on ΔphoP or ΔphoPR system leaving a confusion of the role of PhoR. MPFC and pull-down assay shows interaction of PrrB with PhoP. Then in Fig 4 it is shown that PrrB can phosphorylate PhoP. So my question is why is there no detectable *P-PhoP in ΔphoR::phoP (Fig 2C)?

2: Going by the RNA seq data, the gene list shown in Fig 3A, is it specific for ΔphoR only? What is the number of overlapping genes between ΔphoP and ΔphoR? Also, what number of genes were down regulated in ΔphoR under acidic condition? Authors should also include a heat map of the downregulated genes as shown in fig 3A. Moreover, a Venn diagram showing the list of genes in ΔphoP and ΔphoR along with overlapping gene sets should also be include be better understanding on the control of PhoP and PhoR. This should be followed by a proper discussion.

3: Line 267-268, the authors have mentioned that for their invitro phosphor transfer assay, they have specifically used the N terminal of PhoP and C terminal of PrrB. What was the reason for the authors to select such domain specific region? Will there be any change in results if full length proteins were to be used? Infact, using full length proteins is highly feasible due to significant molecular weight difference between PrrB and PhoP (PrrB is 47.8kd and PhoP is 27.5 kd). If there is any ref paper that has characterized the ATP binding pockets of PrrB that is being followed, then the authors should specify that. Fig 5B should include a control between *P-PhoRC and PhoPN.

4: Fig 5C and 5D really make things confusing. Going by the observation it seems as if PhoR does not control PhoP regulon at acidic stress. This is a shift from the known literature. I would request the authors to reverify this observation once again.

5: Fig7A: please include a better resolution image of the confocal. No differences between the wT-H37Rv and phopR-H37Rv can be concluded from this representative image.

Minor comments:

1: Line 389: please make sure this is 330-fold, the graph looks different.

Reviewer #2: In this manuscript, Singh et al examine the mechanisms underlying the function of the PhoPR two-component system (TCS) in Mycobacterium tuberculosis (Mtb). In particular, they report phosphorylation activity of PhoR induced by acidic pH and high NaCl signals, and the ability of PhoR to also act as a phosphatase. The non-cognate histidine kinase PrrB is further identified as being able to also phosphorylate PhoP. Finally, attenuation in colonization of both macrophages and in vivo in a murine infection model of a ∆phoR mutant Mtb strain is reported.

These studies provide insight into the function of the intensively studied PhoPR TCS, which is known to be critical for Mtb response to acidic pH and high chloride, and for successful host colonization. While attenuation of a ∆phoR mutant Mtb strain has been previously shown, direct demonstration of PhoR phosphorylation activity triggered by bacterial exposure to acidic pH or high NaCl is new, and the phos-tag gel-based experiments in particular present interesting data. The report that PrrB is able to phosphorylate PhoP is also intriguing in examining how different TCSs in Mtb may interact. I have several questions regarding some of the experiments and conclusions drawn, as detailed below.

Major comments:

- Given the authors' pursuit of other kinases that may phosphorylate PhoP based on the continued presence of 35 genes that still exhibit induction upon acidic pH exposure in a ∆phoR mutant (Fig. 3A), it would be helpful if some of those 35 genes are included in the qRT-PCR and ChIP-qPCR experiments shown in Figs. 3B and 3C, to show complementation. This in particular as it appears from Table S3 that there is a range of behavior of these 35 genes in WT Mtb (i.e. it is not that the ∆phoR mutant behaved like WT in expression of these genes; rather, some of these genes were not induced in WT, some were induced more, and some were induced less).

- With the M-PFC assay, are the 3 spots down in each panel supposed to represent triplicate spotting of the same sample? If so, can the authors explain why the results do not replicate, even in the case of the positive control (PhoP-PhoR interaction, lane 4, which also shows very weak positive signal, given the low TRIM concentration used)? Only one spot shows positive signal for the PhoP-PrrB test on the TRIM-containing plates, and the assay results as shown thus do not appear robust. Did the authors try tagging PhoP and/or the various histidine kinases on the opposite terminus to determine which pairings provide the most robust data (i.e. use of the pUAB100 and pUAB200 plasmids of the M-PFC assay system)? The weak signal observed even for the positive control raises questions that other interactions might have been missed in this assay. It is appreciated in the case of the reported interaction between PhoP and PrrB that the co-immunoprecipitation results shown in Fig. 4C do however provide much stronger support.

- prrB knockdown also resulted in increased expression of phoR (Fig. S3A) – given this phenotype and the authors' finding that PhoR has phosphatase activity against phosphorylated PhoP, can the authors ascribe the effects shown in Fig. 5D solely to presumed phosphorylation of PhoP by PrrB? What is effect of prrB knockdown on phoR-dependent genes (genes shown in the qRT-PCR data in Fig. 3B)? The 3 genes selected for analysis in Figs. 5C-5D are all very weakly induced in acidic pH in the ∆phoR mutant. Was there a reason the authors did not examine genes that were more robustly induced in the ∆phoR mutant (e.g. from Fig. 3A: PE20, mvmT, cysK2 etc)?

- Along these lines, the conclusion in lines 312-313 is not explicitly supported by the current data – the authors have shown that PrrB can interact with PhoP and phosphorylate it in vitro, and PrrB is phosphorylated in acidic pH conditions in an intact bacterium, but whether PrrB is indeed then phosphorylating PhoP in the context of an intact wild type bacterium when exposed to acidic pH has not been established. Have the authors tried phos-tag gel assays examining PhoP phosphorylation status in intact WT, phoR knockdown, and prrB knockdown Mtb exposed to acidic pH? In the absence of such direct data, the conclusions made should at minimum be moderated.

Minor comments:

- A ∆phoR mutant Mtb strain has already previously been shown to be attenuated for in vivo colonization in a murine infection model, as indeed the authors briefly reference (current reference 33). This should be better acknowledged however – for example, given this known phenotype, it would seem it is not just the reduced ability of the ∆phoR mutant Mtb to grow in macrophages that would lead one to perform the in vivo infection experiments (lines 385-386).

- PhoPR has previously been reported to be important in Mtb response to high chloride levels, in addition to being critical for the bacterium's acidic pH response (PMID 23592993), which is presumably the reason the authors also tested 250 mM NaCl in their phosphorylation experiments? References should be added as appropriate.

- Mtb PhoR has previously been reported to have autophosphatase activity (PMID 34512602), and possession of phosphatase activity by a histidine kinase in regulation of its cognate response regulator is also known in other TCSs, including in mycobacteria (e.g. SenX3/RegX3 in mycobacteria – PMID 17526710). Acknowledgement and incorporation of these points into the manuscript, perhaps in the discussion section, will be helpful for placing the authors’ work in context.

- The concept of TCS cross-regulation has been studied extensively in other bacteria, and there has been a distinction made between "cross-talk" and "cross-regulation" (see for example PMID 18076326, 24675902). Incorporation of these concepts and comparison to results from other bacterial species would also be helpful in placing the authors' work in context.

- Details for the ChIP-qPCR method currently references a bioRxiv manuscript. Please either cite a peer-reviewed manuscript, or provide full details here.

- Gene names should be all lowercase letters. The authors should also ensure that gene names are accurate (e.g. rv2390c, not rv2390).

Reviewer #3: The manuscrip by Singh et al. attempts to describe the signals sensed by the Mtb PhoP-PhoR two component system. The authors demonstrated that acidic pH and high salt conditions promotes phosphorylation of PhoP and PhoR, which was demonstrated using a phos-tag gels. These gels effectively resolve phosphorylated and unphosphorylated forms of the SK/RR in cell lysates. The authors performed a series of experiments including RNA sequencing, quantitative PCR, and protein protein interaction experiments. A novel finding was demonstrating the interaction between PhoP and PrrB and proving phosphorylation of the former. This was followed by a CRISPRi knockdown experiment and by experiments showing that PhoR functions as a specific phosphatase to maintain the intra-mycobacterial cellular pool of P~PhoP. Lastly the authors perform a series of in vitro and in vivo infections experiments using a Mtb deletion mutant of phoR.

This group has an established track record of studying the PhoP/PhoR two component system. The mechanisms of signal transduction is an important topic in the TB field and authors performed a series of well executed experiments with the proper controls. Hence, I have no concerns regarding scientific rigor and reproducibility. A key finding was demonstrating the interaction between PhoP and PrrB and the subsequent phosphorylation experiments. Overall, it is a well-written manuscrip. However, the have not determined the precise biochemical mechanism whereby PhoR senses acidic pH. In fairness to the authors, this is certainly complex and work will require much more work.

In figure 3, the authors mention that “The data, which represent average of two biological replicates’’. The excel sheet shows average value of the two replicates. It is very difficult to conclude convincingly about the expression levels with just 2 replicates. Why not 3 replicates?

Fig. 7. The region selected for confocal imaging shows only a few Mtb bacilli and the infected cells, the authors should select a region where more infected cells can be seen with and without the lysotracker staining. Also, it should be stated how colocalization has been calculated in Fig. 7B; was performed by manual visualization?

Fig. 7B shows only the three readings (symbols), whereas, in the figure legend, it is mentioned that “% colocalization of auramine labelled strains with Lysotracker was measured in 10 different fields by counting at least 100 infected cells”, whereas in the Methods action it is mentioned that “%co-localization was determined from three biological repeats by analyses of more than 100 bacteria per sample from at least five random fields. Please clarify. Measurement of colocalization using software would be better if possible.

In figure 7 D, did the authors checked for the survival of these mice? Since it was a validation of the previously reported work of Wang et al, additional data of mice survival kinetics would have been more informative. Also, bacterial burden in extra pulmonary organs could also have been assessed.

Line 388-390, it is mentioned that deltaphoR showed an approximately 330-fold lower bacterial burden in mice lungs relative to WT-H37Rv, however, in Fig. 7D it looks like a 2log10 difference which is only 100-fold. Please clarify.

Although the authors have attempted to answer this in the Discussion (lines 455-460); why was PrrB-dependent PhoP phosphorylation not seen in deltaphoR Mtb. The given explanation is not clear. Again, can the authors explain why PhoP phosphorylation was absent in delta-phoR if PrrB can phosphorylate PhoP?

Minor concerns:

Line 87; these references are incorrect, Mtb cannot respire anaerobically. The authors in the PlosOne manuscript only performed transcriptional and proteomic analysis, no respiratory assays were performed. Mtb is an obligate aerobe.

Figure 2B; shouldn’t the blue square be black?

Figure 3A; does the key represent fold/log difference?

Fig. 2B, 3B, 5C-D. The WT-H37Rv bar should have an error bar even if these panels represent the relative expression with the WT-H37Rv.

Fig. legends: please annotated the error bars and the stats used for each panel separately.

In line 1073, “auramine labelled strains” does it mean strains or Mtb?

Line 389, “approximately ~330-fold” please keep either approximately or ~.

**Have all data underlying the figures and results presented in the manuscript been provided?**

Reviewer #1: Yes

Reviewer #2: Yes

Reviewer #3: Yes

PLOS authors have the option to publish the peer review history of their article (what does this mean?). If published, this will include your full peer review and any attached files.

Reviewer #1: **Yes: **Ayan Chatterjee

Reviewer #2: No

Reviewer #3: No

---

## [Decision Letter · Decision Letter 1]

31 Oct 2023

Dear Dr Sarkar,

Thank you very much for submitting your Research Article entitled 'Dual functioning by the PhoR sensor is a key determinant to Mycobacterium tuberculosis virulence' to PLOS Genetics.

The manuscript was fully evaluated at the editorial level and by independent peer reviewers. The reviewers appreciated the attention to an important topic but identified some concerns that we ask you address in a revised manuscript.

We therefore ask you to modify the manuscript according to the review recommendations. Your revisions should address the specific points made by each reviewer.

Yours sincerely,

Danielle A. Garsin

Academic Editor

PLOS Genetics

Sean Crosson

Section Editor

PLOS Genetics

While overall, the reviewers are pleased with this revision, Reviewer #2 has a few remaining comments that I would like to see addressed.

Reviewer's Responses to Questions

**Comments to the Authors:**

Reviewer #1: I thank the Authors for addressing all my comments.

Reviewer #2: The authors have addressed several points of concern in this revised manuscript. A few remaining questions:

Major comments:

- Fig. S2 is very confusing as currently shown, as it is unclear what the Venn diagram in Fig. S2C is showing. Are the numbers in reference to genes that are differentially regulated in ∆phoR or ∆phoP at pH 7 or at acidic pH? The reference from which the authors have taken the ∆phoP data was not done at acidic pH, so presumably this is at pH 7 (or more likely unbuffered)? Yet this figure is called out in a section of the text where the focus is on the response of the bacteria to acidic pH. The text thus does not flow, and it is unclear what this overlap of only 42 genes is supposed to mean, and how this relates to the data described previously in that paragraph. Simply taking the gene list from a different paper is likely somewhat misleading for providing a true comparison, given it is unclear that the conditions under which the RNAseq was done here versus the other paper is comparable. Can the authors please clarify. Also note the figure legend in the supplemental word file does not match that in the pdf file containing all the parts of the manuscript.

- The authors now include new data in Fig. S3 for 2-3 genes that show induction at acidic pH in a ∆phoR mutant. However, no corresponding data for WT and the complemented ∆phoR mutant is shown. This should be included. The strains and RNA would appear to be in hand given the data shown in Fig. 3, so it is unclear why the authors did not include this. As noted in the original review, particularly given the apparent range in behavior of these PhoR-independent, acidic pH-inducible genes in WT Mtb, it is important that complementation of the ∆phoR mutant phenotype be demonstrated here.

Minor comments:

- A suggestion to include Fig. S6 as part of Fig. 7, for ease of reader access to the data.

- Lines 484-486: Phrasing of this sentence is hard to understand. “Despite” does not seem to be the appropriate connecting word here.

Reviewer #3: All my concerns have been addressed.

**Have all data underlying the figures and results presented in the manuscript been provided?**

Reviewer #1: Yes

Reviewer #2: Yes

Reviewer #3: Yes

PLOS authors have the option to publish the peer review history of their article (what does this mean?). If published, this will include your full peer review and any attached files.

Reviewer #1: **Yes: **Dr. Ayan Chatterjee

Reviewer #2: No

Reviewer #3: No

---

## [Editor Report · Decision Letter 2]

16 Nov 2023

Dear Dr Sarkar,

We are pleased to inform you that your manuscript entitled "Dual functioning by the PhoR sensor is a key determinant to Mycobacterium tuberculosis virulence" has been editorially accepted for publication in PLOS Genetics. Congratulations!

Yours sincerely,

Danielle A. Garsin

Academic Editor

PLOS Genetics

Sean Crosson

Section Editor

PLOS Genetics

Comments from the reviewers (if applicable):

**Data Deposition**

http://datadryad.org/submit?journalID=pgenetics&manu=PGENETICS-D-23-00645R2

**Press Queries**

---

## [Editor Report · Acceptance letter]

1 Dec 2023

PGENETICS-D-23-00645R2 

Dual functioning by the PhoR sensor is a key determinant to *Mycobacterium tuberculosis* virulence 

Dear Dr Sarkar, 

We are pleased to inform you that your manuscript entitled "Dual functioning by the PhoR sensor is a key determinant to *Mycobacterium tuberculosis* virulence" has been formally accepted for publication in PLOS Genetics! Your manuscript is now with our production department and you will be notified of the publication date in due course.

With kind regards,

Zsofia Freund

PLOS Genetics

On behalf of:
